# Gene expression composite scores of cellular senescence predict aging health outcomes in the Health and Retirement Study

Qiao Wu [1] ✉, Eric T. Klopack [2], Jung Ki Kim [2], T. Em Arpawong [2], Bharat Thyagarajan [3], Steve Cole[4], Jessica D. Faul[5], Fengxue Zhou[6] & Eileen M. Crimmins [2] ✉

Cellular senescence, a hallmark of aging, can be quantified by the gene expression composite scores for the canonical senescence pathway (CSP), senescence initiating pathway (SIP), senescence response pathway (SRP), a summary of the three, and the SenMayo gene list; however, these have not been probed in representative populations. Using RNA sequencing data from the U.S. representative Health and Retirement Study (HRS) sample (N = 3580), we examine how these composite scores relate to sociobehavioral factors and aging-related outcomes. Senescence scores generally increase with age except for CSP. Higher scores are observed in women and individuals with class II obesity. All scores, except for CSP, are associated with accelerated epigenetic aging, physiological dysregulation, multimorbidity, cognitive decline, and 6-year mortality (all p < 0.05). These associations largely persist after adjustment for DunedinPACE. Our findings suggest that cellular senescence gene expression composite scores capture meaningful variation in aging-related health and complement existing epigenetic aging biomarkers.

Aging is a gradual process across the life cycle; population aging-related health changes can be categorized into sequential dimensions, captured by the morbidity process model[1,2]. Population health changes with aging begin with molecular/cellular-level changes, followed by physiological dysregulation indicated by clinical-level biomarkers, and then by the subsequent diagnosis of health conditions and diseases as well as the development of functional limitations, and then finally death. Cellular senescence is one of the molecular/cellular-level hallmarks of aging that accumulates with advancing age and plays an important pathogenic role in a number of adverse health outcomes[3–7]. Multiple types of cellular/molecular-level damage, including telomere attrition, DNA damage, oxidative stress, mitochondrial dysfunction,

and oncogenic signaling, can potentially induce cellular senescence, changing normal cells to senescent cells. In response to damage, senescent cells stop proliferating and enter into a generally irreversible state of growth arrest[8,9]. In this way, cellular senescence lowers the risk of malignant transformation by preventing the damage from spreading to the next cell generation and is thus a tumor-suppressive mechanism[10]. However, senescent cells are still metabolically active; they release a wide range of pro-inflammatory cytokines, chemokines, proteases, growth factors, and other bioactive molecules to the local microenvironment. Despite the fact that the secretion can help with the clearance of damaged cells and the regeneration of broken tissues, it can have deleterious effects on neighboring cells, contributing to

[1]Max Planck Institute for Human Development, Max Planck Research Group Biosocial–Biology, Social Disparities, and Development, Berlin, Germany. [2]Leonard Davis School of Gerontology, University of Southern California, Los Angeles, CA, USA. [3]Department of Laboratory Medicine and Pathology, University of Minnesota, Minneapolis, MN, USA. [4]Cousins Center for Psychoneuroimmunology, Semel Institute for Neuroscience and Human Behavior, Department of Psychiatry and Biobehavioral Sciences, David Geffen School of Medicine, University of California Los Angeles (UCLA), Los Angeles, CA, USA. [5]Survey Research Center, Institute for Social Research, University of Michigan, Ann Arbor, MI, USA. [6]Berlin School of Public Health, Charité—Universitätsmedizin Berlin, Berlin, Germany. ✉e-mail: wu@mpib-berlin.mpg.de; crimmin@usc.edu

persistent chronic inflammation and progressive fibrosis[3,10,11]. Such proinflammatory secretion is termed Senescence-Associated Secretory Phenotype (SASP), and is thought to mediate downstream aging outcomes[3,10–12]. A high concentration of circulating SASP proteins was found to be associated with pulmonary fibrosis[13], frailty[14], poor physical function[15], and a higher mortality risk[16].

Recent work has suggested a more comprehensive approach to capturing the entire effect of cellular senescence rather than solely relying on SASP—In addition to SASP, other key aspects of cellular senescence include cell cycle arrest (CCA) and macromolecular damage (MD)[8]. To profile these distinct aspects of cellular senescence, Dehkordi et al. developed three lists of genes that are involved in the canonical senescence pathway (CSP), senescence initiating pathway (SIP), and senescence response pathway (SRP) to respectively represent CCA, MD, and SASP[17]. These gene lists were tested and validated in two independent RNA sequencing datasets, and were associated with senescence in previous studies based on various cell and tissue types in human and mouse brains[18–24]. Another gene list reflecting intracellular changes specific to senescent immune cells, termed SenMayo, was recently developed by Saul et al.[25]. SenMayo includes genes involved in CCA, MD, and SASP, and thus has the potential to measure cellular senescence comprehensively. In the current study, we link human blood cell gene expression composite scores based on these gene lists to aging-related health outcomes in a large population-representative sample of older Americans. The representativeness of our sample is important as most work that has been done to date has used samples with limited representativeness as they are either small in size, based on trials, or based on selective electronic health records.

Our interest is in how socio-demographic and behavioral factors get under the skin to affect health outcomes linked to aging[26,27]. Understanding the links between cellular senescence and socio-demographic characteristics like age, sex, socioeconomic status (SES) (e.g., education), and race/ethnicity will provide an indication of how these factors are associated with basic mechanisms of aging, which are prior to well-documented differences in health outcomes. These factors are also linked to health behaviors, which could, in turn, be linked to the MD that triggers cellular senescence; such behaviors include sleep disorders[28–30], obesity and high-fat diet[4,31,32], chronic cigarette smoking[33–35], and alcohol use[36–38]. Evaluating the associations between these factors and cellular senescence at the population level will contribute to our further understanding of how social and behavioral factors get translated to aging biology.

In the current study, we calculated senescence gene expression composite scores in a nationally representative sample of older Americans, based on Dehkordi et al.'s and Saul et al.'s gene lists, and related them to social and behavioral factors hypothesized to be associated with cellular senescence. To understand how cellular senescence is associated with both upstream and downstream aging outcomes, the senescence scores are related to multiple dimensions of aging health—including accelerated epigenetic age, the multi-system physiological dysregulation indicated by accelerated Expanded Biological Age (ExpBioAge), multimorbidity, cognitive function, and mortality. We hypothesize that older age, minority status, low SES, and risky health behaviors or conditions such as smoking, drinking, obesity, and sleep disorders will all be associated with higher levels of cellular senescence. We also hypothesize that the expression of senescence genes will be associated with worse aging-related health outcomes.

## Results

A sample of 3580 Americans 56 years of age and over is included in the current study. The sample has a mean age of 69 years (SD = 9 years), slightly more than half being women (54%), and is predominantly non-Hispanic Whites (78%). As shown in Table 1, the sample has a nationally representative level of education attainment, smoking, alcohol consumption, BMI, and prevalence of comorbid conditions. The average ExpBioAge of the sample is 68 years (SD = 12 years), similar to the average chronological age by design.

Five RNA-based cellular senescence composite scores are used in the current study. The CSP (22 genes), SIP (48 genes), and SRP (44 genes) scores are based on the Dehkordi et al. gene lists. The senescence summary score is a combination of these three scores. The genes in these scores were selected based on brain cells. However, our analysis of data from the Human Protein Atlas (https://www.proteinatlas.org/) shows that cross-gene variations in average expression level in blood correlate well with those observed in brain (in log-transformed nTPM values, CSP: $r = 0.55$; SIP: $r = 0.56$; SRP: $r = 0.52$) (Supplementary Table 1, Supplementary Fig. 1). The SenMayo score is directly based on the Saul et al. gene list, which is developed and validated in bone/bone marrow samples. Among the 125 SenMayo genes, 1 overlaps with the CSP score, 4 overlap with the SIP scores, and 22 overlap with the SRP score. As Fig. 1 shows, all scores are normally distributed. The SIP, SRP, and senescence summary scores are highly correlated ($r = 0.76$–$0.93$), while their correlations with the CSP score are moderate ($r = 0.28$–$0.52$). The senescence summary score is mainly driven by the SIP and SRP scores. Since SenMayo consists primarily of SASP-related genes, it correlates most highly with the SRP score ($r = 0.91$). It is also highly correlated to SIP ($r = 0.80$) and the senescence summary core ($r = 0.88$). The correlations between the senescence scores and the other aging outcome measures (except mortality which is not continuous) are relatively low. The highest correlations are found with the epigenetic clocks.

### Social and behavioral factors predicting the senescence scores

To understand the sociodemographic and behavioral pattern of differences in senescence scores, social and behavioral variables were used together in the same OLS regression model to predict the independent effect of each variable on each score, adjusted for technical covariates (Table 2). Older age groups have significantly higher SRP (aged 65–74: $\beta = 0.06$, $p = 0.002$; aged 75–84: $\beta = 0.13$, $p < 0.001$; aged 85+: $\beta = 0.10$, $p < 0.001$) and SenMayo (aged 65–74: $\beta = 0.04$, $p = 0.043$; aged 75–84: $\beta = 0.11$, $p < 0.001$; aged 85+: $\beta = 0.10$, $p < 0.001$) scores compared to those aged 55–64. For SIP and senescence summary scores, only those aged 75–84 (SIP: $\beta = 0.07$, $p < 0.001$; summary score: $\beta = 0.08$, $p < 0.001$) and those aged 85+ (SIP: $\beta = 0.05$, $p = 0.001$; summary score: $\beta = 0.05$, $p < 0.001$) are significantly higher than the youngest group. The age pattern of CSP is very different from the other scores. All older age groups have significantly lower CSP (aged 65–74: $\beta = -0.05$, $p = 0.019$; aged 75–84: $\beta = -0.09$, $p < 0.001$; aged 85+: $\beta = -0.09$, $p < 0.001$) scores compared to the youngest. Women have higher levels of CSP ($\beta = 0.20$, $p < 0.001$) and senescence summary ($\beta = 0.04$, $p = 0.021$) scores. Compared to non-Hispanic Whites, non-Hispanic Blacks have a higher CSP ($\beta = 0.12$, $p < 0.001$) score but lower SRP ($\beta = -0.07$, $p < 0.001$) and SenMayo ($\beta = -0.07$, $p < 0.001$) scores, and all senescence scores are significantly lower among Hispanics ($\beta$ ranges from $-0.04$ to $-0.06$ depending on the scores, all $p < 0.003$) except for the CSP.

In terms of behavioral factors, higher SIP ($\beta = 0.04$, $p = 0.039$) and lower SenMayo ($\beta = -0.05$, $p = 0.008$) scores are found among those with class I obesity, while those with class II obesity have significantly higher SIP, SRP, and the senescence summary scores ($\beta$ ranges from 0.06 to 0.09 depending on the scores, all $p < 0.002$). A higher level of weekly alcohol consumption is significantly associated with a higher SenMayo score ($\beta = 0.04$, $p = 0.033$).

### The senescence scores and epigenetic aging measures

The associations between each senescence score and each epigenetic aging measure are assessed in OLS regression models, adjusted for all social and behavioral factors and technical covariates. In general, except for the CSP score, all the senescence scores are significantly

## Table 1 | Sample characteristics

| | Mean (SD)/ Percentage | | Mean (SD)/ Percentage |
|---|---|---|---|
| | N = 3580 | | N = 3580 |
| Senescence Scores | | Total Drinks Weekly | 2.9 (6.3) |
| CSP | 0.0 (0.2) | Smoking (Pack Years) | 13.1 (20.7) |
| SIP | 0.0 (0.3) | BMI | 29.4 (6.6) |
| SRP | 0.0 (0.3) | Not Overweight/ Obese | 23.8 |
| Senescence Summary | 0.0 (0.2) | Overweight | 37.4 |
| SenMayo | 0.0 (0.3) | Obesity I | 22.7 |
| Age | 68.6 (9.2) | Obesity II | 16.2 |
| Aged 55–64 | 40.5 | Insomnia Symptoms | 20.4 |
| Aged 65–74 | 35.2 | Cognitive Function | 15.5 (4.3) |
| Aged 75–84 | 17.2 | 6-Year Mortality (N = 3554) | 16.2 |
| Aged 85+ | 7.1 | Multimorbidity | 0.8 (1.0) |
| Female | 54.4 | Diabetes | 25.5 |
| Race Ethnicity | | Cancer | 14.6 |
| Non-Hispanic White | 77.5 | Lung Disease | 11.0 |
| Non-Hispanic Black | 10.2 | Heart Problems | 25.0 |
| Hispanic | 8.9 | Stroke | 8.1 |
| Non-Hispanic Other | 3.3 | ExpBioAge (N = 2660) | 68.2 (11.7) |
| Education | | ExpBioAge AA (N = 2660) | 0.0 (8.1) |
| Years of Education | 13.3 (3.0) | PC GrimAge | 77.3 (8.0) |
| Less than High School | 14.0 | PhenoAge | 56.8 (9.8) |
| High School | 29.9 | DunedinPACE | 1.0 (0.1) |
| Some College | 25.9 | PC GrimAge AA | 0.0 (4.0) |
| College or Higher | 30.2 | PhenoAge AA | 0.0 (6.8) |

Italic values indicate mean and standard deviations for continuous variables.

*CSP* canonical senescence pathway score, *SIP* senescence initiating pathway score, *SRP* senescence response pathway score, *Senescence Summary* senescence summary score, *Sen-Mayo* SenMayo score, *ExpBioAge* expanded biological age, *AA* age acceleration, *PC* principal component.

associated with faster epigenetic age acceleration. Using the epigenetic aging pace algorithm, DunedinPACE, as an example (Fig. 2A, Supplementary Table 2), the SIP, SRP, senescence summary, and Sen-Mayo scores are all linked to faster pace of aging (SIP: $\beta = 0.26$, $p < 0.001$; SRP: $\beta = 0.19$, $p < 0.001$; Summary score: $\beta = 0.24$, $p < 0.001$; SenMayo: $\beta = 0.16$, $p < 0.001$). The other two epigenetic aging measures have similar associations with the senescence scores (Supplementary Table 2).

### The senescence scores and downstream aging-related health outcomes

The associations between each senescence score and each aging-related health outcome among ExpBioAge acceleration, multimorbidity, and cognitive function are assessed in OLS regression models, and the associations between senescence scores and 6-year mortality are assessed in logistic regression models, adjusted for all social and behavioral factors and technical covariates (Fig. 2B, C, Supplementary Table 2). For models using ExpBioAge acceleration as the outcome, the sample size is 2660 due to missing biomarkers. For the models using 6-year mortality as the outcome, the sample size is 3554 due to missing vital status.

Except for the CSP score, all senescence scores are significantly associated with higher odds of 6-year mortality (SIP: OR = 1.67; SRP:

OR = 1.38; summary score: OR = 1.52; SenMayo: OR = 1.32; All $p < 0.001$), worse cognitive function (SIP: $\beta = -0.07$; SRP: $\beta = -0.05$; summary score: $\beta = -0.07$; SenMayo: $\beta = -0.05$; all $p < 0.05$), multi-morbidity (SIP: $\beta = 0.15$; SRP: $\beta = 0.09$; summary score: $\beta = 0.12$; SenMayo: $\beta = 0.07$; all $p < 0.001$), and faster ExpBioAge acceleration (SIP: $\beta = 0.23$; SRP: $\beta = 0.16$; summary score: $\beta = 0.20$; SenMayo: $\beta = 0.13$; all $p < 0.001$). Overall, for the SIP, SRP, senescence summary, and SenMayo scores, the size of the beta coefficients predicting biological age is the largest, followed by those predicting multimorbidity, and then those predicting cognitive functioning.

Finally, DunedinPACE is included in all the models as a covariate to see whether senescence scores explain additional variation to that explained by epigenetic age in aging-related health outcomes (Supplementary Table 3). After adjusting for DunedinPACE, all previously observed significant associations remain at least marginally significant (all $p < 0.10$). Specifically, SIP (OR = 1.43, $p < 0.001$), SRP (OR = 1.22, $p = 0.004$), senescence summary (OR = 1.30, $p = 0.001$), and SenMayo (OR = 1.18, $p = 0.021$) scores remain significantly associated with mortality. For cognitive function, the coefficients of SIP ($\beta = -0.06$, $p = 0.006$) and senescence summary ($\beta = -0.05$, $p = 0.008$) scores remain significant. For multimorbidity, the coefficients of SIP ($\beta = 0.09$, $p < 0.001$), SRP ($\beta = 0.04$, $p = 0.029$) and senescence summary ($\beta = 0.07$, $p = 0.004$) scores remain significant. For ExpBioAge acceleration, the coefficients of SIP ($\beta = 0.16$, $p < 0.001$), SRP ($\beta = 0.11$, $p < 0.001$), senescence summary ($\beta = 0.13$, $p < 0.001$), and SenMayo ($\beta = 0.08$, $p = 0.002$) scores remain significant.

## Discussion

The current study examines how five blood gene expression composite scores of cellular senescence link to social-behavioral characteristics and aging-related health outcomes in a nationally representative sample of older Americans. The CSP, SIP, and SRP scores correspond to three dimensions of cellular senescence (CCA, MD, and SASP, respectively), and the senescence summary score is a combination of the three. SenMayo is a separate measure of cellular senescence, combining mainly SASP genes with some CCA- and MD-related genes. As expected, we find that older individuals generally have higher senescence scores in peripheral blood, especially for SASP (indicated by both the SRP and SenMayo scores). However, this was not true for all scores as we found the CSP score decreases with age. Our study provides preliminary human evidence on the sex difference in cellular senescence at the population level. In addition, people with class II obesity generally have a higher level of cellular senescence.

We also find that cellular senescence links to aging-related biomarkers and health outcomes. The senescence scores are significantly associated with epigenetic aging measures, and downstream aging-related health outcomes including accelerated biological age, multi-morbidity, cognitive function, and 6-year mortality. These associations are driven by the SIP and SRP but not by the CSP. If we think of these various dimensions of health as a process[1,2], the beta coefficients of the senescence scores are generally greater in the models measuring upstream aging measures and attenuate gradually as the outcome of the model moves downstream, possibly because those downstream outcomes are more distal and are potentially influenced by additional factors. When both senescence scores and DNAm indicators of pace of epigenetic aging are used to predict downstream aging-related outcomes, the SIP, SRP, senescence summary, and SenMayo scores continue to significantly predict mortality, cognitive function, multimorbidity, and biological age (SRP is only marginally associated with cognitive function; SenMayo is only marginally associated with multimorbidity and cognitive function). This indicates that the senescence scores explain additional variation in health outcomes beyond what is explained by current epigenetic clocks.

Though most senescence scores are higher among older age groups, the expression level of CSP genes is significantly lower

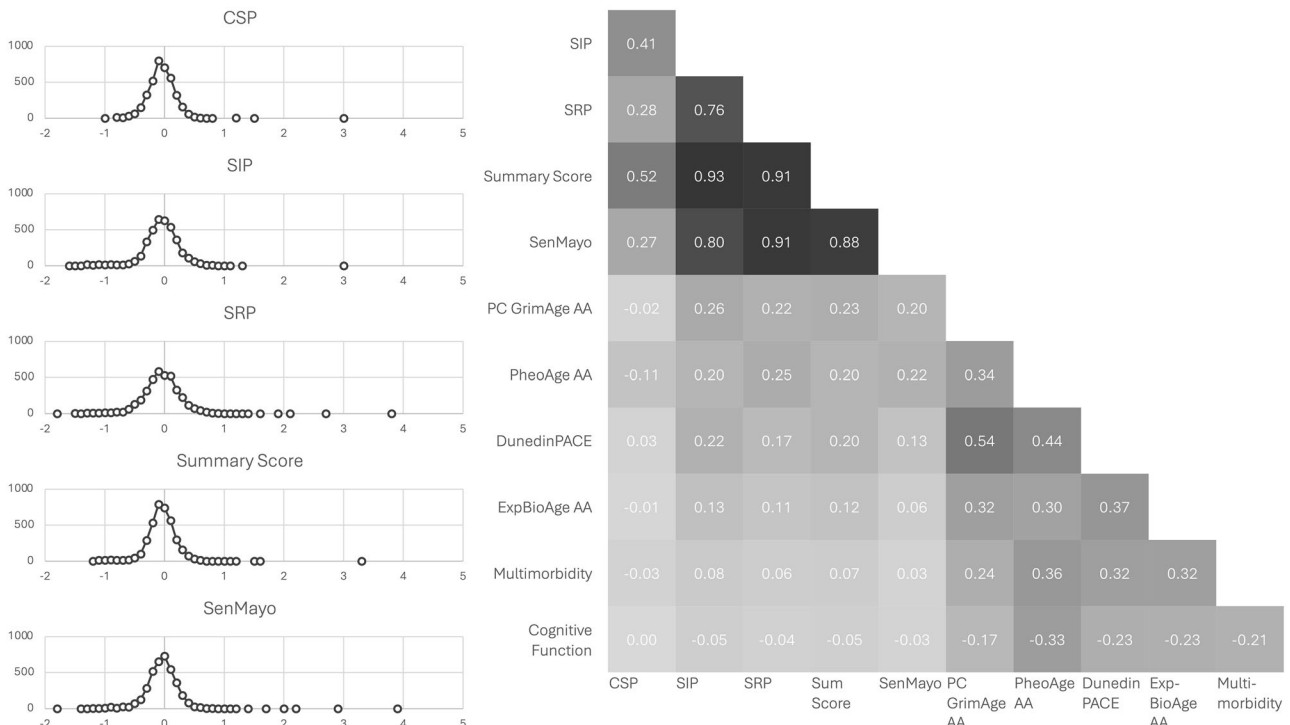

**Fig. 1 | The distribution and correlation matrix of the cellular senescence scores.** CSP canonical senescence pathway Score, SIP senescence initiating pathway score, SRP senescence response pathway score, Sum Score senescence summary score, SenMayo SenMayo Score, ExpBioAge Expanded Biological Age, AA age acceleration, PC Principal Component. Pearson correlation coefficients (*r*) are reported in the correlation matrix.

(Table 2). In addition, unlike other senescence scores, the CSP score is not significantly associated with aging-related outcomes (Fig. 2). These results imply that the genes included in the CSP list might capture an aspect of senescence which is not entirely detrimental. In fact, the capacity to induce CCA in response to stress and damage could be important and beneficial under certain circumstances, for example to prevent the proliferation of damaged cells and malignant transformation[10]. Choi and Lim found the induction of p21 is compromised in aging cells (the corresponding gene CDKN1A is included in the CSP list used in the current study)[39]. Previous studies also suggest that although CCA is considered a hallmark of cellular senescence, cell-cycle re-entry can occur under certain adverse circumstances particularly in tumor cells[8]. Collectively, these previous works and our findings suggest that the effective activation of CCA could be compromised while aging. Relevantly, female sex is associated with higher CSP and senescence summary scores (Table 2). Since both SIP and SRP are not significantly associated with female sex, the elevated senescence summary score among females is likely mainly driven by CSP. Given the well-known lifespan advantage of females, the sex difference in CSP could be interpreted as the reduced capacity of CCA among males.

In our sample, minority status is associated with lower levels of cellular senescence (i.e., lower SRP score for non-Hispanic Blacks; lower SIP, SRP, senescence summary, and SenMayo scores for Hispanics), which is opposite to our expectation. People who belong to racial/ethnic minority groups generally have more adverse exposures (e.g., environmental, chemical, and psychological) and fewer protective resources (e.g., occupational, social, and healthcare) throughout their lifespan, and hence are hypothesized to have a higher level of senescence. The proportions in the youngest age group (aged 55–64) among racial/ethnic minority groups are all significantly larger than that among non-Hispanic Whites. The proportions of aged 75–84 among racial/ethnic minority groups are all significantly lower than that among non-Hispanic Whites, indicating differential survival across

racial/ethnic groups (Supplementary Table 4). One explanation is that, in our analytical sample, people from racial/ethnic minority groups who survive to older age are more resilient and are healthier in the aspects that are relevant to the level of cellular senescence. However, such differential patterns of survival across racial/ethnic groups should be considered as the consequence of complex socioeconomic disparity rather than the influence of distinct underlying biological mechanisms.

Assumptions made about vital status could influence our results on mortality. If no contact is made with a respondent in 2022, the respondent is presumed to be alive. In order to test the effect of this assumption, we excluded the 404 respondents who were presumed alive in our mortality models in a sensitivity analysis. After exclusion, the results remain very similar both in terms of ORs and *p* values, with and without adjustment for DunedinPACE (Supplementary Table 5).

Gene expression profiles can differ across cell types; therefore, we conducted sensitivity analyses to account for between-person variability in immune cell composition. A common approach is to adjust for DNA methylation-based estimates of major immune cell subsets[40,41]. However, the HRS includes flow cytometry data, which directly provides the percentages of these major immune cell subsets, including granulocytes, natural killer cells, B cells, CD4+ T cells, CD8+ T cells, and monocytes. When we controlled for these cell-type proportions, most of the age effects on senescence scores were no longer significant (Supplementary Table 6), and certain associations between senescence scores and aging-related health outcomes were attenuated or lost significance (Supplementary Tables 7, 8). One plausible explanation for this attenuation is that immune cell distribution may lie on the causal pathway between the expression of cellular-senescence-related genes and the other variables of interest. For example, cellular senescence, in part through the SASP, can alter immune cell composition, which may subsequently affect aging-related health outcomes. Because our primary objective was to identify whether a link exists between cellular senescence and these variables—rather than to

**Table 2 | Results of ordinary least squares regression models predicting the senescence scores (N = 3580)**

| N = 3580 | CCA | | MD | | SASP | | Sum Score | | SenMayo | |
|---|---|---|---|---|---|---|---|---|---|---|
| | beta | p | beta | p | beta | p | beta | p | beta | p |
| Age: Ref—Aged 55–64 | | | | | | | | | | |
| Aged 65–74 | −0.05* | 0.019 | 0.02 | 0.252 | 0.06** | 0.002 | 0.03 | 0.123 | 0.04* | 0.043 |
| Aged 75–84 | −0.09*** | 0.000 | 0.07*** | 0.000 | 0.13*** | 0.000 | 0.08*** | 0.000 | 0.11*** | 0.000 |
| Aged 85+ | −0.09*** | 0.000 | 0.05*** | 0.001 | 0.10*** | 0.000 | 0.05*** | 0.000 | 0.10*** | 0.000 |
| Female | 0.20*** | 0.000 | −0.03 | 0.063 | 0.02 | 0.176 | 0.04* | 0.021 | 0.01 | 0.613 |
| RE: Ref—Non-Hispanic White | | | | | | | | | | |
| Non-Hispanic Black | 0.12*** | 0.000 | −0.02 | 0.300 | −0.07*** | 0.000 | −0.02 | 0.213 | −0.07*** | 0.000 |
| Hispanic | −0.01 | 0.450 | −0.04** | 0.003 | −0.06*** | 0.001 | −0.05*** | 0.001 | −0.05** | 0.002 |
| Non-Hispanic Other | 0.01 | 0.805 | −0.02 | 0.183 | −0.04* | 0.011 | −0.03 | 0.050 | −0.03 | 0.078 |
| Education: Ref—Less than High School | | | | | | | | | | |
| High School | −0.02 | 0.367 | 0.00 | 0.874 | 0.02 | 0.368 | 0.00 | 0.883 | 0.03 | 0.262 |
| Some College | 0.00 | 0.852 | −0.02 | 0.447 | 0.00 | 0.932 | −0.01 | 0.651 | 0.01 | 0.573 |
| College and Higher | −0.02 | 0.506 | −0.04 | 0.116 | −0.01 | 0.842 | −0.03 | 0.254 | 0.02 | 0.515 |
| BMI: Ref—Normal | | | | | | | | | | |
| Overweight | 0.01 | 0.732 | 0.02 | 0.368 | −0.01 | 0.757 | 0.01 | 0.623 | −0.03 | 0.109 |
| Obese I | 0.02 | 0.511 | 0.04* | 0.039 | −0.02 | 0.383 | 0.02 | 0.383 | −0.05** | 0.008 |
| Obese II | 0.02 | 0.292 | 0.09*** | 0.000 | 0.06** | 0.002 | 0.08*** | 0.000 | 0.00 | 0.924 |
| Cumulative Packs Smoked | 0.02 | 0.185 | 0.02 | 0.182 | −0.01 | 0.650 | 0.01 | 0.443 | −0.01 | 0.576 |
| Total Drinks Weekly | 0.02 | 0.207 | 0.01 | 0.621 | 0.03 | 0.142 | 0.02 | 0.254 | 0.04* | 0.033 |
| Insomnia Symptoms | 0.01 | 0.576 | 0.02 | 0.266 | 0.00 | 0.904 | 0.01 | 0.527 | −0.01 | 0.732 |
| Adjusted R2 | 0.20 | | 0.40 | | 0.25 | | 0.34 | | 0.29 | |

The model is adjusted for batch/plate. Standardized coefficients are reported. All social and behavioral variables are included in the same multivariate regression models where senescence scores are dependent variables.

*CSP* canonical senescence pathway score, *SIP* senescence initiating pathway score, *SRP* senescence response pathway score, *Sum Score* senescence summary score, *SenMayo* SenMayo score.

*p < 0.05, **p < 0.01, ***p < 0.001.

illustrate specific pathways or separate direct and indirect effects—the unadjusted model results remain most relevant to our aims. Nevertheless, these sensitivity analyses underscore the importance of clarifying how cellular senescence is triggered and how it influences health outcomes through other mechanisms such as immunosenescence.

Since the analyses in the current study were hypothesis-driven, and for each socio-behavioral factor and for each aging outcome only four models were conducted corresponding to the four senescence scores, multiple testing corrections were not globally applied to our main analyses. However, given the total number of models, the Benjamini-Hochberg False Discovery Rate (FDR) adjustment was performed as a sensitivity test. For the models predicting senescence scores, FDR adjustment was applied across models with different dependent variables (senescence scores) (Supplementary Table 9). Since most of the previously observed associations had fairly high significance levels, the FDR adjustment did not change our interpretation or conclusions. Similarly, for the models using senescence scores to predict aging-related health outcomes, FDR adjustment was applied across models sharing the same outcome (Supplementary Tables 10, 11). To three digits after the decimal point, the FDR adjusted p values did not change compared to the original p values. Again, the correction did not change our interpretation or conclusion.

Second- and third-generation epigenetic clocks are more appropriate indicators of health-related biological aging because they focus on DNAm patterns associated with aging phenotypes, health risks, mortality, or health indicators instead of only chronological age. Empirically, second- and third-generation clocks have demonstrated better performance in terms of reflecting social determinants of health (e.g., SES) and predicting aging-related health outcomes[26,42,43]. We

limited the total number of clocks included in the analysis to avoid creating an unnecessarily large number of models. However, we have additionally examined the association between senescence scores and the two first-generation clocks—PC Horvath2 age acceleration and PC Hannam age acceleration[44]. As expected, the associations of senescence scores with PCHorvath2AA and PCHannumAA are generally weaker than the associations with the second- and third-generation clocks (Supplementary Table 12).

The current study has limitations. First, although the gene expression composite scores applied here were constructed using pre-defined gene lists from the literature and not derived or optimized using the current dataset, it remains important to rigorously evaluate their predictive performance on the selected outcomes, ideally in an independent cohort comparable to the HRS. However, the HRS is unique in its combination of a large, population-representative sample, high-quality multi-omics data, and comprehensive aging phenotyping. At present, accessing a comparable dataset for external validation is challenging. As an alternative, we conducted 5-fold cross-validation (CV) for all models linking senescence scores to aging outcomes. The average CV $R^2$ values, full-sample $R^2$ values, RMSE, and MAE are reported in Supplementary Table 13. In general, CV $R^2$ values were slightly lower than those from the full-sample models, indicating that model explanatory power may decline modestly when applied to new populations. The drop in $R^2$ values and the ratio of RMSE and MAE were particularly notable in models predicting 6-year mortality, suggesting potential overfitting in that context. As more population-level multi-omics datasets become available, further assessment of the generalizability and predictive performance of these composite scores will be possible.

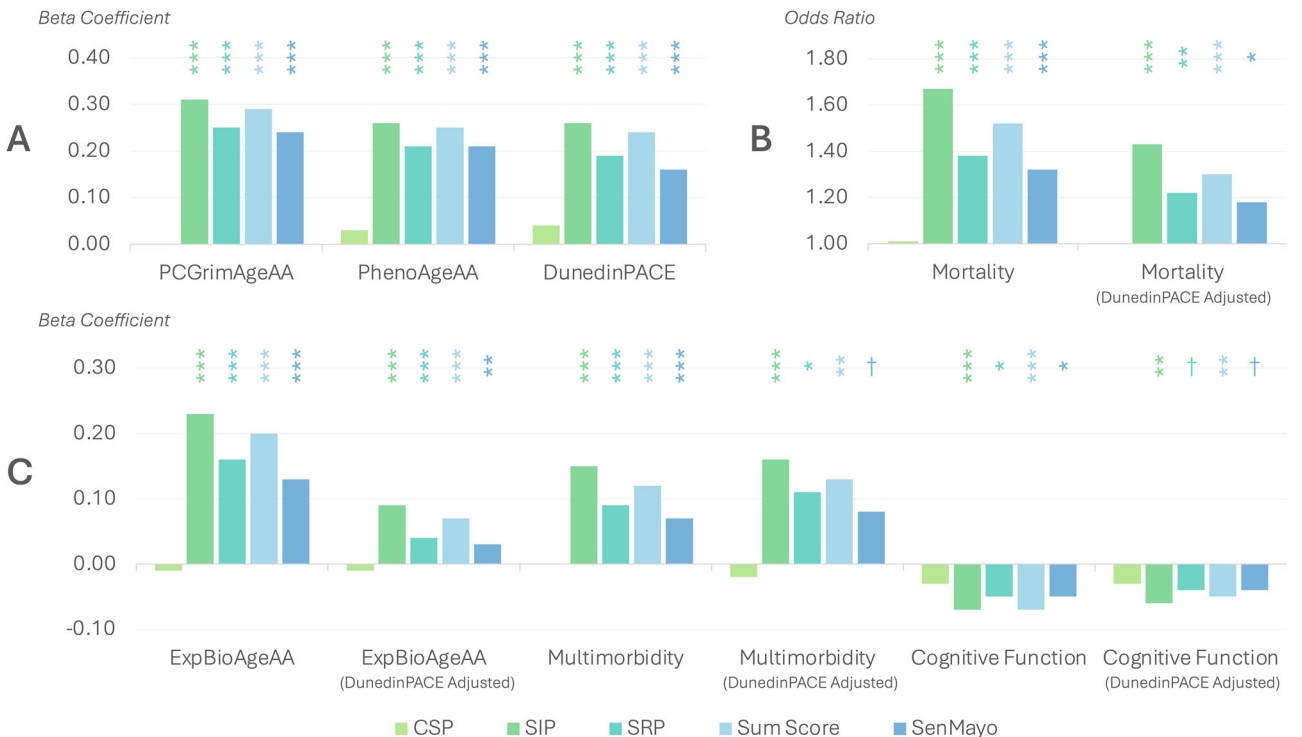

**Fig. 2 | The associations between five cellular senescence scores and multiple aging-related health outcomes.** †$p < 0.1$ *$p < 0.05$, **$p < 0.01$, ***$p < 0.001$. CSP canonical senescence pathway Score, SIP senescence initiating pathway score, SRP senescence response pathway score, Sum Score Senescence Summary Score, SenMayo SenMayo Score, ExpBioAge Expanded Biological Age, AA age acceleration, PC Principal Component. All models are adjusted for all covariates (age, sex, race/ethnicity, education, BMI categories, smoking, drinking, and insomnia symptoms) and batch/plate. **A** Shows the associations between the senescence scores and epigenetic aging measures. **B** Shows the associations between the senescence scores and 6-year mortality. **C** shows the associations between senescence scores and other downstream health outcomes. When mortality is the outcome, odds ratios are reported, and senescence scores and DunedinPACE are standardized for comparison. Standardized coefficients are reported for the other outcomes. For the model predicting mortality, $N = 3554$; for the model predicting ExpBioAgeAA, $N = 2660$; for all other models, $N = 3580$. Source data used to generate this figure can be seen in Supplementary Tables 2, 3.

Second, some aging-related health outcomes used in the current study rely on self or survivor reports, including multimorbidity and 6-year mortality. Although self-reports are commonly used in population surveys, future studies with a focus on diseases/conditions or mortality could leverage information sources such as health records. Third, representativeness is a strength of our sample, but it also reflects the sex and age structure of the population, especially the sex differences in survival to older ages. Specifically, the proportion of female participants in our study steadily increases with advancing age (e.g., 52.6% among those aged 55–64, 52.9% among those aged 65–74, 57.0% among those aged 75–84, and 65.9% among those aged 85+). Thus, a sample with more balanced sex ratios and a more evenly distributed age pattern could be more suitable for future studies that have a focus on the sex and age pattern of cellular senescence. Finally, this is a cross-sectional study, so no conclusions can be drawn regarding the causal role of senescence in any of the aging-related outcomes analyzed here.

Although our senescence scores were not developed to maximize predictive accuracy for clinical applications, their sensitivity to social and behavioral factors and strong associations with multiple dimensions of aging suggest significant potential for future research. These scores could serve as valuable tools for identifying mechanisms by which social and behavioral exposures influence aging biology. These scores may also aid in preclinical and clinical evaluations of interventions aimed at reducing the burden of cellular senescence. While clinical research plays a critical role in aging research, population-level survey data provide unique value by offering insights that are difficult to obtain in smaller, controlled clinical settings. Population-level

studies, such as the Health and Retirement Study (HRS), enable the investigation of aging across diverse subpopulations, capturing the impact of social, economic, and behavioral factors on health outcomes. These data sources are also critical for understanding the interplay between biological and non-biological determinants of health, identifying disparities, and informing public health interventions. The HRS, with its nationally representative sample of older Americans, with extensive RNA-seq data and comprehensive collection of social, economic, behavioral, and health information, is a uniquely powerful resource for testing similar hypotheses and advancing translational research in aging. Maximizing prediction is not the aim of the current study, but future studies dedicated to that specific mission could consider combining the gene lists used in this study, or building on them to develop new algorithms.

We conclude that whole blood RNAseq data provide a useful approach for measuring cellular senescence in population health studies. RNA-based cellular senescence scores are associated with demographic, socioeconomic, and behavioral factors, and they add health-predictive information above and beyond that available from measures derived from DNA methylation. RNA-based senescence measures also capture multiple dimensions of aging and add to our understanding of mechanisms promoting aging-related physiological changes and diseases.

## Methods

This secondary data analysis received IRB approval (UP-18-00229) from the Human Research Protection Program, University of Southern California.

## Data and sample

Our study sample comes from the HRS, a longitudinal study of US adults older than age 50. In 2016, a representative subsample of HRS Venous Blood Sample (VBS) participants was selected for innovative assays reflecting cellular-/molecular-level mechanisms of aging, including RNA sequencing. After the quality control process, a total of 3738 respondents have valid gene expression values. All participants with RNA data also have valid epigenetic aging measures based on DNA methylation. Our main analytical sample consists of 3580 respondents who further have complete information on sociodemographic characteristics, behavioral factors, and multimorbidity. The models using 6-year mortality as the outcome are based on a subsample of 3554 respondents who have non-missing vital status at the 2022 follow-up. The models using ExpBioAge as the outcome are based on a subsample of 2660 respondents who further have non-missing corresponding biomarkers. The measurement occasions for data collection can be seen in Supplementary Table 14.

## RNA extraction

RNA was extracted from whole blood stored in Paxgene tubes by using the Paxgene Blood miRNA Kit. Extracted RNA is then stored at −80 °C until further analysis. Total RNA isolates were quantified using a fluorimetric RiboGreen assay. Total RNA samples were treated with the Globin-Zero Gold rRNA Removal Kit (Illumina Inc.) to deplete ribosomal RNA and globin prior to creating sequencing libraries using Illumina's stranded mRNA Sample Preparation kit (Cat. # RS-122-2101). One microgram of total RNA was oligo-dT purified using oligo-dT coated magnetic beads, fragmented, and then reverse transcribed into cDNA, fragmented, blunt-ended, and ligated to indexed (barcoded) adaptors and amplified using 15 cycles of PCR. Indexed libraries were then normalized, pooled and size selected to 320 bp ±5% using Caliper's XT instrument.

## RNA sequencing

Samples were sequenced using $2 \times 50$ bp paired-end reads to a minimum of 20 million reads per sample on NovaSeq at the University of Minnesota Genomics Center. All samples were processed through the HRS RNAseq quality control (QC) analysis pipeline at the University of Minnesota. This is an extended version of the TopMed/GTEX analysis pipeline (https://github.com/broadinstitute/gtex-pipeline/blob/master/TOPMed_RNAseq_pipeline.md). The STAR aligner was used for alignment of the sequence reads to the GRCh38 human reference genome along with GENCODE 30 annotations. All quality control analyses were performed using an updated version of RNASeQC 2.3.4 and estimated quality control metrics to obtain the final data. The read counts from each sample were combined into a count file.

## Normalization/transformation

We used edgeR calcNormFactors() function and used RLE (relative log expression) normalization to account for compositional differences between the libraries. RLE is the scaling factor method; where the median library is calculated from the geometric mean of all columns and the median ratio of each sample to the median library is taken as the scale factor. We then used cpm() function in edgeR on the normalized DGEList object to estimate the log2 counts-per-million (log2cpm) with a prior.count = 2.

## Gene expression composite scores

Five gene expression composite scores are used in the current study— The CSP score (based on 22 genes), the SIP score (48 genes), the SRP score (44 genes), the senescence summary score (112 genes) which is a combination of the previous three (gene lists are from Dehkordi et al.[17]), and the SenMayo score (125 genes, gene list is from Saul et al.[25]). Specifically, genes involved in CSP code the proteins that are important for establishing cell cycle arrest/withdrawal (CCA); SIP

genes code the proteins primarily involved in DNA repair, oxidative stress, and telomere shortening, indicating the overall level of MD; SRP genes code pro-inflammatory cytokines and chemokines, growth modulators, angiogenic factors, and matrix metalloproteinases that are involved in SASP (Supplementary Tables 1, 15). *IGFBP7* and *AKT1* genes are in both SIP and SRP scores, and they are only used once in the senescence summary score. The SenMayo gene list reflects intracellular changes specific to senescent cells and mainly consists of SASP-related genes but with an inclusion of some CCA- and MD-related genes as well. Specifically, among the 125 SenMayo genes, 1 overlaps with the CSP score, 4 overlap with the SIP score, and 22 overlap with the SRP score (Supplementary Table 15). Each score is calculated as the mean of the z-score standardized log2 transformation of the normalized transcript abundance value of the corresponding genes. A higher value of the score indicates a higher level of cellular senescence.

Specifically, for respondent $i$, the $Score_i$ is calculated using the equation below.

$$Score_i = \frac{\sum_g^G \left( \frac{Log_2 CPM(g)_i - mean(Log_2 CPM(g))}{sd(\log_2 CPM(g))} \right)}{G}$$

The subscript g indicates individual genes, and G indicates the total number of genes. CPM stands for count per million, which is the transcript counts of gene g per million total human transcriptome-mapped RNA sequencing reads, indicating the normalized expression level of gene g. Since the average expression level can vary substantially across genes, a log2 transformation is applied to the CPM values for each gene to prevent the results from being dominated by a few highly expressed genes. Since expression level can be heteroscedastic, a z-score standardization is applied to the log2 CPM values for each gene to prevent the results from being dominated by a few highly variable genes. Finally, the scores average the standardized log2 transformed expression level of all genes.

## Outcome measures

**Mortality.** The mortality information is derived from the vital status of the respondent attached to the 2022 HRS interview, and thus measures 6-year mortality. This information results from reports of survivors or previously designated contacts. In 2022, categories of vital status include (1) alive at this wave, (2) presumed alive as of this wave (no contact was made), (3) known deceased as of this wave, and (4) known deceased as of prior wave. In the main analysis, respondents who were known deceased as of wave 2022 and prior waves (category 3 and 4) were coded as deceased and those known or presumed alive (categories 1 and 2) were coded as alive. As a sensitivity analysis, the 404 respondents who were presumed alive (category 2) were excluded from the sample used for models predicting mortality.

**Cognitive function.** Cognitive function is measured using a modified version of the Telephone Interview for Cognitive Status (TICS-m) in 2016. A cognitive function score ranging from 0 to 27 was generated by summing up the imputed TICS-m items provided in the HRS data, with higher values indicating better cognitive function.

**Multimorbidity.** The multimorbidity measure is based on information collected in 2016. It is the number of self-reports of five physician-diagnosed health conditions: (1) diabetes or high blood sugar; (2) cancer or a malignant tumor of any kind except skin cancer; (3) chronic lung disease except asthma such as chronic bronchitis or emphysema; (4) heart attack, coronary heart disease, angina, congestive heart failure, or other heart problems; and (5) stroke or transient ischemic attack[26]. The multimorbidity measure ranges from 0–5.

**Expanded biological age.** Biological age is measured using the expanded biological age (ExpBioAge)[42] based on 22 clinical-level biomarkers. It indicates the general level of physiological dysregulation and explains multimorbidity well at the population level. Among the 22 biomarkers, two of them are from HRS physical measurement (systolic blood pressure and peak flow), and one is dried blood spots biomarker collection (HbA1c). For these three markers, data from a random half of the sample were collected in 2014, and those from the other half were collected in 2016. To measure ExpBioAge beyond the effect of chronological age, ExpBioAge acceleration is used in the current study. This is calculated using the residual that results from regressing biological age on chronological age. Since biological age (acceleration) is measured in years, it can be interpreted as age acceleration or deceleration compared to chronological age.

**Epigenetic age.** Based on DNA methylation data from the 2016 venous blood collection of HRS, epigenetic age is estimated by 3 DNA methylation clocks—GrimAge[45] and PhenoAge[46], both trained on aging-related health outcomes, and DunedinPACE[47], trained on within-individual change across 19 biological indicators. To bolster the reliability of and reduce the influence of technical noise on the epigenetic clock algorithms, the principal component (PC) version[44] of GrimAge is used in the current study. Similarly to biological age, epigenetic age acceleration (AA) reflects faster or slower aging according to the epigenetic clocks[26]. As the outcomes of our models, PC GrimAge AA and PhenoAge AA are represented in years. DunedinPACE captures the pace of epigenetic aging. It cannot be interpreted in years; instead, it indicates the AA per year. Results for PC GrimAge are shown in the main text and results for PhenoAge and DunedinPACE are shown in the supplementary material in order to reduce the content of tables and figures and because they performed similarly.

### Sociodemographic and behavioral measures
Sociodemographic and behavioral measures include age, self-reported sex, race/ethnicity, education, alcohol consumption, smoking, body mass index (BMI) categories, and sleep disorder. Age is categorized into 4 groups: 56–64, 65–74, 75–84, and 85 and older (85+). Racial/ethnic groups include non-Hispanic White, non-Hispanic Black, Hispanic, and non-Hispanic Others. Education is classified as less than high school, high school, some college, and college degree or higher. Alcohol consumption is measured by self-reported total weekly drinks, which is the product of the average daily number of drinks and the total days of drinking per week. Smoking is measured by lifetime pack-years, which is the product of the average daily number of cigarette packs smoked and the lifetime years of smoking[48]. Sleep disorder is indicated by the presence of insomnia symptoms, which is a combined measure of sleep disturbance with non-restorative sleep[49]. All the sociodemographic and behavioral measures are collected in 2016, except for BMI. BMI is calculated using measured height and weight, but when physical measures are not available, self-reported values are used. Then, respondents are categorized into not overweight/obese ($BMI < 25$), overweight ($25 \leq BMI < 30$), obesity I ($30 \leq BMI < 35$), and obesity II ($BMI \geq 35$). For height and weight, data from a random half of the sample were collected in 2014, and those from the other half were collected in 2016.

### Technical covariates
Dummy codes indicating 46 batch plates were included as technical covariates in all models.

### Statistical analysis
First, to understand the sociodemographic and behavioral pattern of senescence scores, social and behavioral variables were included together in the same OLS regression model to predict each senescence score, adjusted for technical covariates. Then, the associations between each senescence score and epigenetic aging measures are assessed in OLS regression models, adjusted for all social and behavioral factors and technical covariates. Then, the associations between each senescence score and each aging-related health outcome among ExpBioAge acceleration, multimorbidity, and cognitive function are assessed in OLS regression models, and the associations between senescence scores and 6-year mortality are assessed in logistic regression models, adjusted for all social and behavioral factors and technical covariates. Finally, DunedinPACE is included in all the models using the senescence score to predict aging-related outcomes as a covariate to see whether senescence scores explain additional variation to that explained by epigenetic aging in these outcomes. Survey weights for the HRS VBS innovative sub-sample (2016 VBS weight for analyses including DNA methylation and epigenetic clocks, telomeres, and homocysteine) are used to adjust for initial sample selection and differential consent and completion. All analyses are performed using Stata version 18.

### Reporting summary
Further information on research design is available in the Nature Portfolio Reporting Summary linked to this article.

### Data availability
Much of the Health and Retirement Study (HRS) data can be accessed via the University of Michigan site using the online application system (https://hrs.isr.umich.edu/data-products). Sensitive health data from HRS is only available under restricted access: Venous Blood Study (VBS) and Biomarker Data used in the current study can be accessed using the HRS sensitive health data online application system (https://hrsdata.isr.umich.edu/data-products/sensitive-health). HRS epigenetic and transcriptomic raw data files can be accessed through The National Institute on Aging Genetics of Alzheimer's Diseases Data Storage Site (NIAGADS) upon application (https://hrs.isr.umich.edu/data-products/genetic-data). More information on the NIAGADS application can be found on its official website (https://niagads.scrollhelp.site/support/application-instructions). Questions for the HRS team can be sent to hrsquestions@umich.edu or submitted through the online contact system (https://hrs.isr.umich.edu/help). Source data are provided with this paper.

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

## Acknowledgements

This study is supported by the National Institute on Aging (P30 AG017265). The Health and Retirement Study is supported by the National Institute on Aging (U01 AG009740) and the Social Security Administration. Q.W. is supported by the Max Planck Society.

## Author contributions

Q.W. contributed to the study design, conducted the statistical analyses, and drafted the manuscript. E.T.K., J.K.K., and T.E.A. contributed to data cleaning and statistical analyses. B.T., S.C., and J.D.F. assisted with data preparation and analysis. F.Z. performed supplementary analyses and supported data visualization. E.M.C. guided the study design, analytical strategy, and manuscript development. All authors contributed to the interpretation of results and helped critically revise the manuscript.

## Funding

## Competing interests

The authors declare no competing interests.
