## [Peer Review file · Nature Communications]

Gene Expression Composite Scores of Cellular Senescence Predict Aging Health Outcomes in the Health and Retirement Study

Corresponding Author: Dr Qiao Wu

Version 0:

Reviewer comments:

Reviewer #1

(Remarks to the Author)

Comments for NCOMMS-24-73623-T

RNA-based Indicators of Cellular Senescence Predict Aging Health Outcomes in the Health and Retirement Study

#####

The study has great data types that are not always available at the same time (RNAseq, DNAm, cell counts, other phenotypic data and the different poor outcomes) and intriguing, timely relevant research objective.

I have some suggestions that need to be considered prior to publishing. My main concerns are related epigenetic ages (and the aging rate) and cell subtypes (in the blood samples).

-The clocks are developed using data driven approach (i.e. black box analysis). Currently, we really don't know how the clocks tick i.e. what is the internal mechanism of the clocks and what exactly makes the clock values to behave as true biological age values (they do have properties of biological age indicators!). Therefore, my first question is that what does it actually mean to adjust for the clock values when the internal mechanisms of the clocks are unknown?

-My second issue is about the clocks and also blood cell types. Very recent evidence have shown that the epigenetic ages (calculated using different clock algorithms) are different across blood cell subtypes. In general, recent evidence suggest that during human adulthood, while aging, blood cell types with 'young' epigenetic ages become less abundant in blood circulation while 'old' cells become more common. Please, see e.g. publications: Zhang Z, et al. 2023, <https://doi.org/10.1111/accel.14071> ; Martila, S et al. 2024. <https://doi.org/10.1007/s11357-024-01287-w>). These findings suggest the clock values can be linked to immunosenescence/cellular senescence. Thus, it may be also so that adjusting for epigenetic ages can also adjust for blood cell subtypes (to some unknown extent). I think this adjustment by clock values needs more consideration. However, it is interesting that the main findings hold even when adjusting for the clocks (BUT NOT when adjusting for some main blood cell types). One side note is that I suggest GrimAge is a bit difficult clock when studying smoking. GrimAge was trained to detect smokers (among other things). I suggest to highlight some other clock than GrimAge. And to think about the first point I wrote above (=unknown mechanisms).

-Third concern is about sensitivity analyses stating in line 288-298: '... most of the age effects on senescence scores become insignificant (Supplemental Table 15). Some of the associations between senescence scores and age-related health outcomes are attenuated or no longer significant (Supplemental Tables 15-23). This is hard to interpret as the distribution of immune cells could theoretically lie in the causal pathway between the expression of cellular senescence and other variables being tested, the inclusion of cell composition could be considered overadjustment.' Why including results that can not be interpreted? More importantly, as blood cell subtypes have different gene expression profiles and in this study, gene expression is used to measure cellular senescence, it is indeed very likely that adjusting for some cell subtypes can attenuate the associations heavily. However, I am not sure if the cell types in this analysis were the best choices. Why those cell types were used in the analysis? Do they represent immunosenescence/senescence in full? If not, what does that mean i.e. how this adjustment can be interpreted. I think this needs more explanation/consideration.

- Fourth point: could you explain vital status in more details? I interpret it was not register-based but it is not easy to follow.
- Fifth point: in methods, please, specify the statistical program/programming language for all sections.
- Sixth point: I suggest to write a bit more specific conclusion-statement in abstract instead of: 'RNA-based senescence scores enhance our understanding of aging mechanisms related to physiological decline and health outcomes.'
- Seventh point: could you explain why multiple testing correction was not needed? Some evaluation for that (maybe not to manuscript but reply to the reviewers).
- Last point: Because these data types are so expensive, it can be understood easily why only one data set was used. However, it might be good if some crossvalidation analysis (as sensitivity analysis) is made. It would highlight if the findings are robust.

Reviewer #2

(Remarks to the Author)

The paper by Wu et al. analyzes RNA/transcriptomics-based markers, namely CCA, MD, SASP, a summary senescence score, and SenMayo in the Health and Retirement Study and associates them with sociodemographic and behavioral factors and several age-related outcomes, including mortality, multimorbidity, biological age acceleration and epigenetic age acceleration. The find that the senescence scores mostly increased with age, were higher in women and individuals with class II obesity and associated with epigenetic aging, accelerated biological age, multimorbidity and mortality. While I feel that the field definitely needs such new markers of biological aging that may provide a more comprehensive picture of aging and help us better understand the process, the work unfortunately suffers from several very fundamental flaws, such as a lack of correction for multiple testing and replication in an independent sample. There is also no description of the statistical analyses (an entire of two sentences of which the other says which program was used and the other refers to survey weights, which tells nothing about the analyses), so it is impossible to evaluate the statistical methods, let alone model parametrization and assumptions. I also feel that the paper suffers from lack of organization and clarity in the hypothesis and rationale. I hope my comments are helpful.

- In the title and throughout the text, I feel it is misleading to refer the scores to as "RNA-based Indicators of Cellular Senescence" as the scores are essentially gene expression profiles/transcript abundancies. More appropriate term, as used by the authors in the methods, is composite gene expression scores.
- There are already several established transcript signatures associated with aging and/or aging-related traits, so it would be relevant to demonstrate to which extent the senescence-associated transcripts are represented in the existing signatures/scores. This would also help illustrate the added value of the senescence-related transcripts.
- The introduction is somewhat unfocused and long, presenting as list of topics that tap the tested exposures and outcomes, without a clear rationale or grounded hypothesis that would provide a foundation for the present work.
- To meet the aim stated by the authors "To understand how these composite scores are then associated with both potential upstream and downstream aging outcomes, they are related to multiple dimensions of aging – including accelerated epigenetic age, the multi-system physiological dysregulation indicated by accelerated Expanded Biological Age (ExpBioAge), multimorbidity, and mortality." this work would have to be better structured, more rigorous and provide a solid rationale for the approach. Now it unfortunately feels like seemingly random testing of all possible associations with variables that were available in the sample.
- Related to the above, if the authors were to aim at comprehensive characterization of the senescence-associated gene expression scores, the paper would need a description of the statistical analyses, model specifications, tested assumptions, correction for multiple testing and most importantly, replication in an independent sample. All these are unfortunately lacking now. The only cues to the models can be obtained from the table headers and footnotes, which is certainly not enough to evaluate what has been done and how. As I understand from the tables, it seems that the models were only adjusted for technical covariates, so the associations (called "regressions" by the authors) are essentially bivariate correlations that are extremely prone to confounding. Therefore, the true nature and relevance of the results is hard to judge.
- As said above, I was very surprised to find that the data analysis section is only two sentences. Please provide a fully developed description of statistical analysis, which models were used, how the model assumptions were tested and met, how the models were parameterized and what was the rationale to define something as an exposure and something as an outcome. This is particularly relevant for analyses on sociodemographic and economic factors and BMI. Which way around you expect the effect-outcome - sequence to be? Please also consider confounding and adjust for it.
- A basic principle and ground rule of analyses like these, where a great number of associations is tested, is, after carefully adjusting for confounding, to perform correction for multiple testing, using FDR or Bonferroni correction. The associations that pass such correction are then usually replicated in an independent sample to determine robustness and generalizability. While I understand that not all associations found here can be tested in an independent sample, such testing should be performed at least for the main findings. I am also certain that after appropriate adjustment for confounding and multiple testing, the significant associations will become much fewer and easier to replicate. There are several publicly available datasets with transcriptomics or RNA-seq data, some including more detailed sample and covariate data linked, so these

could serve as starting point for the replication. If not sufficient, perhaps collaboration with consortia or research groups in possession of such data is an option? Last resort option is to perform cross-validation.

- Last sentence of the first paragraph in the result says "The average biological age of the sample is 68 years (SD=12 years), similar to the average chronological age by design." Biological age can be measured in several different ways and using various scales and scores (epigenetic clocks are also biological ages!), so the reader would appreciate more detailed clarification in this context as to what is meant by biological age here (albeit the intro and methods provide some cues). I assume it is the ExpBioAge that you are referring to? If yes, I don't think it is appropriate to call the ExpBioAge as "the" biological age, rather use the exact term and indicate which level of biological aging this taps into (cellular, organ, system-wide etc.)

- Why only PhenoAge, GrimAge and DunedinPACE were considered among the epigenetic clocks? While older and trained only on chronological age, the Horvath and Hannum clocks would surely be worthwhile testing too.

- Also puzzles me a bit as to why the PC version was used only for GrimAge but not for the other clocks? Usually one uses the clock versions consistently and if inconsistencies are observed between the original and the PC versions, they are reported and discussed.

- Please clarify the assessment of the multimorbidity score, were these ever diagnoses or was a specific look-back window applies? Since this is based on self-report data, has any validation been made against diagnosis codes or established indices like CCI?

In the discussion, the authors acknowledge that adjusting for cell composition attenuated most of the associations to null and interpret this to be a result of "overadjustment". I find the interpretation very unusual as in such cases the attenuation usually indicates that it is the cell types that are (independently) associated with the outcomes, not the transcript levels. It is a very standard methodology for example in DNA methylation analysis to adjust for blood cell composition, as failing to do so can lead to spurious findings for the same reason. Please consider revising the interpretation.

- In the light of the above, the limitations-paragraph in the discussion is very much underdeveloped, but this can be remedied by addressing the main methodological and approach-related issues.

- Would be very helpful if the authors provided a figure indicating the measurement occasions for collecting the data and obtaining the different measures.

Minor:

- In the abstract please specify what SenMayo refers to
- In the table footnotes, I believe you are referring to batch with "patch"?

Reviewer #3

(Remarks to the Author)

Review of the Manuscript entitled "RNA-based Indicators of Cellular Senescence Predict Aging Health 2 Outcomes in the Health and Retirement Study" by Wu Q and coauthors

Briefly, using cross-sectional data from a large and nationally representative subsample of men and women, aged 56 years or older, from the U.S. Health and Retirement Study with available RNA sequencing data, the authors investigated the association between cellular senescence and age-related health outcomes. Five RNA-based cellular senescence scores (CCA, MD, SASP, a summary senescence score, and SenMayo) were calculated. Age related health outcomes included epigenetic age, the multi-system physiological dysregulation indicated by accelerated Expanded Biological Age (ExpBioAge), multimorbidity, and mortality. As result, senescence scores increased with age, except for CCA, which decreased. Women and individuals with class II obesity exhibited higher senescence levels. All senescence scores, except CCA, were significantly associated with epigenetic aging, accelerated biological age, multimorbidity, and 6- year mortality. Noteworthy, these associations remained significant after adjusting for GrimAge.

It is well-recognized that cellular senescence is a key mechanism in the patho-physiology of aging. As acknowledged in the introduction of the manuscript, previous studies showed a significant association between cellular senescence and age-related chronic conditions. Indeed, a point of strength of the analysis by Wu and coauthors is the large dimension of the sample population. Moreover, they used a comprehensive approach to measure cellular senescence. Regarding methodology, authors used an appropriate study design, valid and reliable measures and a correct analytic approach. Results are valid and robust, providing a significant contribution to the existing literature on this topic.

However, my comments are:

1. Contrary to what expected, CCA score was found to decrease with age and was not associated with age-related health outcomes. How this finding could be interpreted? I suggest to add a paragraph about this in the discussion.
2. Multimorbidity included only 5 conditions: diabetes, cancer, chronic lung disease, heart disease and stroke. I would suggest to include a measure of cognitive decline among the age-related health outcomes to investigate whether cellular senescence is associated also with impair cognitive function in the sample population.

Reviewer #4

(Remarks to the Author)

This descriptive study provides novel and valuable evidence that "senescence scores" derived from whole blood RNA sequencing (RNAseq) data can predict age-related health outcomes, including 6-year mortality, and demonstrate associations with socioeconomic and behavioral factors.

While descriptive, the study addresses an important gap in the field and highlights the potential utility of senescence biomarkers that can be obtained from blood from living humans in aging research. The manuscript is noteworthy, particularly because it validates the use of four out of five senescence scores in human whole blood cells—a setting where their utility has not been previously well-established. The findings could have significant implications for future human studies requiring reliable aging biomarkers, as well as preclinical research using animal models to evaluate geroprotective interventions for healthspan extension.

The manuscript presents a good case for publication and will be of interest to a broad audience of researchers working on aging and its associated health outcomes. However, I have some questions and suggestions.

Major Comments:

1. Renaming of Senescence Score Gene Lists:

The authors renamed three gene lists derived from Dehkordi et al. (2021): "Canonical Senescence Pathway" to "Cell Cycle Arrest (CCA)," "Senescence Initiators" to "Macromolecular Damage (MD)," and "Senescence Responses" to "Senescence-Associated Secretory Phenotype (SASP)." While the renaming is logical and perhaps enhances interpretability, it would be beneficial if the authors could explicitly clarify the rationale for these changes in the text. A brief explanation in the methods or discussion section would provide transparency and perhaps help readers understand the biological implications from the named gene lists better.

2. Sex Distribution Across Age Groups:

The study benefits from an approximately equal proportion of sexes overall, but the distribution of males and females across the four age groups is unclear. Given the significant sex differences in aging and life expectancy (e.g., 76.2 years for males and 81.1 years for females in 2016 in United States, according to Natl Vital Stat Rep 2019 May;68(4):1-66), it is likely that later age groups (e.g., 75–84 and 85+) include a higher proportion of females. Providing sex-specific proportions for each age group would offer valuable context and help interpret the results.

3. Sex-Specific Analysis:

As discussed briefly in 'Discussion' section in the manuscript, sex differences in aging are well-documented. Women tend to experience earlier onset of age-related conditions (particularly those driven by senescence and inflammation) but outlive men globally, regardless of socioeconomic status. This suggests that the aging process, including senescence progression, may differ between sexes. For instance, females may exhibit earlier associations between senescence scores and age-related conditions, while males could "catch up" at later ages. A bioinformatics study has shown there might be such a possibility: Ezra et al., doi: <https://doi.org/10.1101/2023.02.27.530179>.

To explore this, it would be insightful to test associations between senescence scores and age-related outcomes in a sex-specific and age-specific manner, rather than grouping all age ranges together with sex included as a covariate. It is possible that weak associations observed at younger ages become stronger at older ages because of converging senescence score trajectories between sexes. Even though this is a cross-sectional study and sample sizes for the 85+ group may be limited, the data could potentially reveal intriguing sex-specific patterns in the relationship between senescence scores and aging related health outcomes.

4. Ethnicity Distribution Across Sex and Age Groups:

The manuscript also examines differences across ethnic groups, but the distribution of sex and age within each ethnicity is not provided. Since both age (and sex) are key variables influencing senescence and health outcomes, understanding their distribution across the four ethnic groups is critical for interpreting the findings. It would be helpful to clarify whether differences in senescence scores between ethnicities might be partly driven by imbalances in sex or age distributions.

5. Combination of MD and SenMayo Scores:

Among the senescence scores, the MD (Macromolecular Damage) score appears to be the strongest predictor of age-related health outcomes. Would it be worthwhile to explore whether combining MD with the SenMayo score strengthens the associations? For example, if the combination improves predictive power, it may indicate complementary contributions of these gene lists. Conversely, if the association weakens, it could suggest that SenMayo includes genes less relevant to these specific outcomes, even if the gene list captures senescence itself. Testing such combinations could provide valuable insights into the relative contributions of each score and guide future refinements in senescence biomarkers.

Additional Comments:

• Clarity of Age-Related Associations:

The authors observe that associations between senescence scores and age-related outcomes are weaker in younger age groups and become stronger with increasing age. While this observation is consistent with the progressive nature of senescence, further discussion on the potential biological mechanisms driving this trend might enhance the manuscript. For example, could this pattern reflect accumulating senescent cells with age or the increasing impact of senescence-related inflammation and macromolecular damage in older individuals?

• Future Applications in Translational Research:

The authors could expand on the potential translational applications of their findings. For example, how might these senescence scores be used in clinical settings to assess patient risk for age-related diseases or mortality? Additionally, the manuscript could discuss the implications for preclinical research for translation, such as how these scores might aid in evaluating geroprotective drugs or other interventions aimed at reducing senescence burden and/or increasing healthspan.

• Methodological Clarifications:

Were any steps taken to address potential confounding effects of medications, polypharmacy or comorbidities on senescence scores?

Version 1:

Reviewer comments:

Reviewer #1

(Remarks to the Author)

Thank you for the nice revision. No further comments.

Reviewer #2

(Remarks to the Author)

The manuscript has improved for several parts in the revision. However, my foremost concern, replication in an independent sample or cross-validation as a bare minimum, remains unaddressed.

Regarding replication, the authors state that "We agree that cross-validation or external validation is critical for studies aimed at developing new biomarkers. However, we did not develop the biomarkers used in the current study. Instead we applied existing biomarkers, which have been validated in their own ways before (our study is a validation of those measures in some ways). These measures are not all available in a harmonized representative older sample that can be used for replication".

As I specified in my initial comment (and as acknowledged by the authors themselves), replication (or cross-validation) is typically considered an essential component of biomarker validation, rather than a separate or optional downstream step. While the biomarkers applied in the current study have been previously developed, they have not been validated (replicated) in the context of the predictions presented here. The authors should therefore consider to seek replication in an independent sample or, where that is not feasible, to implement cross-validation strategies within the current dataset.

Reviewer #3

(Remarks to the Author)

I think that the authors well-addressed the comments raised by reviewers and that the revised version of the manuscript has increased its quality

Reviewer #4

(Remarks to the Author)

I thank the authors for their responses and performing extra data analysis. I am happy about their suggested changes, and the explanations/arguments on the matters which had not been changed.

Version 2:

Reviewer comments:

Reviewer #2

(Remarks to the Author)

The inclusion of cross-validation analysis is a significant improvement. However, the results are not adequately addressed—they are only presented in the supplementary materials and not discussed at all in the main text. This omission is problematic, particularly given the cross-validation results for 6-year mortality. The CV R^2 values are notably lower than the main model R^2 , with drops exceeding 0.10, which commonly suggest overfitting. This may indicate that the model is overly complex or capturing noise rather than true signal. Similarly, the RMSE/MAE ratios exceeding 1.5 point to substantial variance in prediction errors, possibly due to outliers. These issues should be openly discussed in the manuscript to provide a transparent interpretation of the findings and their implications.

Moreover, the authors state that "The gene expression composite scores we applied were constructed based on existing gene lists from the literature and were not derived or optimized using the current dataset. No training, model fitting, or parameter tuning was performed in this study." This statement is not entirely accurate. While the individual genes may have been previously associated with aging-related outcomes, the composite signature developed here is new. Whenever a new predictive combination is created—whether labelled a "signature" or "profile"—its performance needs to be rigorously assessed. The best practice is validation in an independent cohort, or if that is not feasible, thorough internal validation such as cross-validation.

If the authors wish to maintain the claim that no training or model fitting was performed, they should acknowledge that this limits the robustness of their approach.

Finally, the absence of replication in an independent cohort should be clearly stated as a limitation of the study.

Version 3:

Reviewer comments:

Reviewer #2

(Remarks to the Author)

The authors have addressed my comments -- good work.

Reviews Comment Response

I am Qiao Wu, the first and corresponding author of this manuscript. I would like to sincerely apologize for the extended delay in our revision process. Unfortunately, I recently underwent a major surgery and required a period of recovery, which significantly impacted my ability to work during that time. I am now fully recovered and able to resume my responsibilities. I truly appreciate the editor's and reviewers' patience and understanding throughout this period. Thank you very much!

Reviewer #1 (Remarks to the Author):

The study has great data types that are not always available at the same time (RNAseq, DNAm, cell counts, other phenotypic data and the different poor outcomes) and intriguing, timely relevant research objective. I have some suggestions that need to be considered prior to publishing. My main concerns are related epigenetic ages (and the aging rate) and cell subtypes (in the blood samples).

1. The clocks are developed using data driven approach (i.e. black box analysis). Currently, we really don't know how the clocks tick i.e. what is the internal mechanism of the clocks and what exactly makes the clock values to behave as true biological age values (they do have properties of biological age indicators!). Therefore, my first question is that what does it actually mean to adjust for the clock values when the internal mechanisms of the clocks are unknown?

We agree that the exact mechanisms underlying epigenetic clocks remain to be fully elucidated, but think it is important to point out that their strong associations with aging-related health outcomes have been well-documented in the literature. As such, they are the most widely used and validated indicators of biological aging. However, it is important to acknowledge that epigenetic alterations represent only one of the hallmarks of aging (i.e., the underlying molecular/cellular processes of aging, López-Otín et al., 2023), and DNA methylation is just one type of epigenetic modification. Consequently, DNAm clocks are unlikely to provide a comprehensive measure of biological aging. Biological aging measures based on other hallmarks of aging may capture additional variation in aging-related outcomes, as they might reflect underlying mechanisms that DNAm clocks do not account for. In this study, we adjusted for epigenetic clocks in our models to evaluate whether senescence scores add explanatory power beyond what is captured by DNAm clocks. Since the senescence scores remain significant after the clock adjustment, we believe that senescence scores hold significant potential, as they may reflect biological aging mechanisms that DNAm clocks do not capture.

2. My second issue is about the clocks and also blood cell types. Very recent evidence have shown that the epigenetic ages (calculated using different clock algorithms) are different across blood cell subtypes. In general, recent evidence suggest that during human adulthood, while aging, blood cell types with 'young' epigenetic ages become less abundant in blood circulation while 'old' cells become more common. Please, see e.g. publications: Zhang Z, et al. 2023, <https://doi.org/10.1111/ace.14071> ; Marttila, S et al. 2024. <https://doi.org/10.1007/s11357-024-01287-w>). These findings suggest the clock values can be linked to immunosenescence/cellular senescence. Thus, it may be also so that adjusting for epigenetic ages can also adjust for blood cell subtypes (to some unknown extent). I think this adjustment by clock values needs more consideration. However, it is

interesting that the main findings hold even when adjusting for the clocks (BUT NOT when adjusting for some main blood cell types). One side note is that I suggest GrimAge is a bit difficult clock when studying smoking. GrimAge was trained to detect smokers (among other things). I suggest to highlight some other clock than GrimAge. And to think about the first point I wrote above (=unknown mechanisms).

In terms of blood cell subtypes:

We agree with the reviewer that epigenetic clock values can be influenced by blood cell distributions. As noted, recent studies have demonstrated that different blood cell subtypes exhibit distinct epigenetic ages and that age-related shifts in cell composition (Marttila et al., 2024; Zhang et al., 2024), suggesting that epigenetic clocks may partially reflect immunosenescence and cellular senescence. Consequently, adjusting for epigenetic clocks in our models may account for blood cell subtypes to some extent. However, we do not believe that this issue compromises our conclusions. Our analysis adjusting for blood cell distributions indicates that some associations between senescence scores and age-related health outcomes are no longer significant, so cell adjustment attenuates the main effects. If adjusting for DNAm clocks inadvertently introduces the effect of cell type composition, it would be reasonable to expect that this adjustment would similarly attenuate the main associations. The fact that the main findings remain robust even after adjusting for DNAm clocks implies that the senescence scores capture additional explanatory power beyond what is reflected by the clocks.

In terms of the selection of GrimAge for adjustment in our models:

We acknowledge that GrimAge should be used cautiously when assessing the relationship between smoking and biological aging, as it was partially trained on smoking history. Though, in our current models adjusted for GrimAge, smoking is neither an outcome nor a covariate, which reduces concerns about potential biases. We selected GrimAge as the DNAm clock for adjustment because of its relatively superior performance in predicting multimorbidity and mortality in the HRS sample (Crimmins et al., 2024; Faul et al., 2023). But since DunedinPACE has fairly similar performance in predicting multimorbidity and mortality in previous work, to address the potential concerns regarding GrimAge, we now use DunedinPACE for adjustment in our models. After adjusting for DunedinPACE, all previously observed significant associations remain at least marginally significant (All $p < 0.10$). This does not substantially change our conclusions. We have revised results, discussions, and methods sections accordingly.

The Associations between Four Cellular Senescence Scores and Multiple Aging-Related Health Outcomes

Note

† p<0.1 * p<0.05, ** p<0.01, *** p<0.001

CSP – Canonical Senescence Pathway Score; SIP – Senescence Initiating Pathway Score; SRP – Senescence Response Pathway Score; Sum Score – Senescence Summary Score; SenMayo – SenMayo Score; ExpBioAge – Expanded Biological Age; AA – Age Acceleration; PC – Principal Component.

All models are adjusted for all covariates (age, sex, race/ethnicity, education, BMI categories, smoking, drinking, insomnia symptoms) and batch/plate.

Panel A shows the associations between the senescence scores and epigenetic aging measures. Panel B shows the associations between the senescence scores and 6-year mortality. Panel C shows the associations between senescence scores and other downstream health outcomes. When mortality is the outcome, odds ratios are reported, and senescence scores and DunedinPACE are standardized for comparison. Standardized coefficients are reported for the other outcomes. For the model predicting mortality, N=3,554; for the model predicting ExpBioAgeAA, N=2,660; for all other models, N=3,580.

3. Third concern is about sensitivity analyses stating in line 288-298: ‘... most of the age effects on senescence scores become insignificant (Supplemental Table 15). Some of the associations between senescence scores and age-related health outcomes are attenuated or no longer significant (Supplemental Tables 15-23). This is hard to interpret as the distribution of immune cells could theoretically lie in the causal pathway between the expression of cellular senescence and other variables being tested, the inclusion of cell composition could be considered overadjustment.’ Why including results that can not be interpreted? More importantly, as blood cell subtypes have different gene expression profiles and in this study, gene expression is used to measure cellular senescence, it is indeed very likely that adjusting for some cell subtypes can attenuate the associations heavily. However, I am not sure if the cell types in this analysis were the best choices. Why those cell types were used in the analysis? Do they represent immunosenescence/senescence in full? If not, what does that mean i.e. how this adjustment can be interpreted. I think this needs more explanation/consideration.

We agree that results that cannot be fully interpreted should not be reported as main findings. However, cell type adjustment is a widely used approach in both DNA methylation and RNA studies, and we feel it is important to include these results to provide transparency and address potential concerns from readers familiar with this methodology. (In our case, we believe that adjusting for blood cell composition may represent overadjustment, as gene expression is used to measure cellular senescence, and blood cell subtypes have distinct gene

expression profiles. If blood cell composition lies on the causal pathway between cellular senescence and the health outcomes being studied, such adjustment could attenuate meaningful associations by removing part of the effect we aim to capture.) However, we presented them rather than omitting these findings to ensure scientific transparency. We think it is important to acknowledge that the adjustment attenuates the main effect, and we are not hiding this from the readers. So we frame this as a sensitivity check and mention this in the discussion section.

We selected the cell types based on the Reinius and Salas estimations of cell distribution, which is a commonly used approach for cell distribution adjustment in the literature of biological aging. These two estimations generally rest on using DNA methylation data to estimate the distribution of granulocytes, natural killer cells, B cells, CD4 cells, CD8 cells, and monocytes. However, since we have percentages of these cells from flow cytometry, we believe that directly using these percentages is a strength of our study.

The revised paragraph in the discussion is shown below:

Gene expression profiles can differ across cell types; therefore, we conducted sensitivity analyses to account for between-person variability in immune cell composition. A common approach is to adjust for DNA methylation-based estimates of major immune cell subsets (Reinius et al. 2012; Salas et al. 2018). However, the HRS includes flow cytometry data, which directly provides the percentages of these major immune cell subsets, including granulocytes, natural killer cells, B cells, CD4+ T cells, CD8+ T cells, and monocytes. When we controlled for these cell-type proportions, most of the age effects on senescence scores were no longer significant (Supplemental Table 6), and certain associations between senescence scores and aging-related health outcomes were attenuated or lost significance (Supplemental Tables 7 and 8). One plausible explanation for this attenuation is that immune cell distribution may lie on the causal pathway between the expression of cellular-senescence-related genes and the other variables of interest. For example, cellular senescence, in part through the SASP, can alter immune cell composition, which may subsequently affect aging-related health outcomes. Because our primary objective was to identify whether a link exists between cellular senescence and these variables—rather than to illustrate specific pathways or separate direct and indirect effects—the unadjusted model results remain most relevant to our aims. Nevertheless, these sensitivity analyses underscore the importance of clarifying how cellular senescence is triggered and how it influences health outcomes through other mechanisms such as immunosenescence.

4. Fourth point: could you explain vital status in more details? I interpret it was not register-based but it is not easy to follow.

We thank the reviewer for bringing this up. We briefly mentioned how we coded the mortality variable in the discussion section when introducing the corresponding sensitivity analysis. We now provide more details in the method section. Vital status is not register-based but mortality ascertainment in the HRS results from reports of survivors and contacts whose names were provided at earlier interviews. It is effectively complete as documented in this report (validating mortality ascertainment in the HRS: <https://hrs.isr.umich.edu/publications/biblio/9022>)

The revised part of the methods section reads as follows:

Mortality – The mortality information is derived from the vital status of the respondent attached to the 2022 HRS interview, and thus measures 6-year mortality. This information results from reports of survivors or previously designated contacts. In 2022, categories of vital status include (1) alive at this wave, (2) presumed alive as of this wave (no contact was made), (3) known deceased as of this wave, and (4) known deceased as of prior wave. In the main analysis, respondents who were known deceased as of wave 2022 and prior waves (category 3 & 4) were coded as deceased and those known or presumed alive (categories 1 & 2) were coded as alive. As a sensitivity analysis, the 404 respondents who were presumed alive (category 2) were excluded from the sample used for models predicting mortality.

5. Fifth point: in methods, please, specify the statistical program/programming language for all sections.

This information can be found at the end of the Data Analysis section. We specified that all analyses are performed using Stata version 18.

6. Sixth point: I suggest to write a bit more specific conclusion-statement in abstract instead of: 'RNA-based senescence scores enhance our understanding of aging mechanisms related to physiological decline and health outcomes.'

We now reworded the end of the abstract. Other parts of the abstract have also been revised based on other comments we received. The revised abstract can be seen below:

Cellular senescence, a hallmark of aging, can be quantified through the expression levels of genes involved in the canonical senescence pathway (CSP), senescence initiating pathway (SIP), and senescence response pathway (SRP). How cellular senescence links to sociodemographic characteristics, behavioral factors, and aging-related health outcomes in representative populations remains unknown. Using a nationally representative subsample from the U.S. Health and Retirement Study with RNA sequencing data, we calculated five cellular senescence gene expression composite scores: CSP, SIP, SRP, a summary senescence score, and SenMayo (based on the SenMayo gene list). Linear regression models assessed their associations with sociodemographic and behavioral factors (N=3,580), as well as aging-related health outcomes, including mortality (N=3,554), cognitive function (N=3,580), multimorbidity (N=3,580), biological age acceleration (N=2,660), and epigenetic age acceleration (N=3,580). Senescence scores increased with age ($\beta=0.04-0.13$, all $p<0.043$), except for CSP, which decreased ($\beta=-0.05$ to -0.09 , all $p<0.019$). Women ($\beta=0.04$, $p=0.021$) and individuals with class II obesity ($\beta=0.08$, $p<0.001$) exhibited higher senescence levels. All senescence scores, except CSP, were significantly associated with epigenetic aging, accelerated biological age, multimorbidity, cognitive function, and 6-year mortality (all $p<0.05$). These associations remained at least marginally significant after adjusting for DunedinPACE. Our results indicate that cellular senescence gene expression composite scores add to the explanation of health outcomes by epigenetic mechanisms.

7. Seventh point: could you explain why multiple testing correction was not needed? Some evaluation for that (maybe not to manuscript but reply to the reviewers).

The following paragraph and corresponding tables (below) have been added to the discussion section:

Since the analyses in the current study were hypothesis-driven, and for each socio-behavioral factor and for each aging outcome only four models were conducted corresponding to the four senescence scores, multiple testing corrections were not globally applied to our main analyses. However, given the total number of models, the Benjamini-Hochberg False Discovery Rate (FDR) adjustment was performed as a sensitivity test. For the models predicting senescence scores, FDR adjustment was applied across models with different dependent variables (senescence scores). Since most of the previously observed associations had fairly high significance levels, the FDR adjustment did not change our interpretation or conclusions. Similarly, for the models using senescence scores to predict aging-related health outcomes, FDR adjustment was applied across models sharing the same outcome. To three digits after the decimal point, the FDR adjusted p values did not change compared to the original p values. Again, the correction did not change our interpretation or conclusion.

Predicting Senescence Scores - FDR Adjusted Version

N=3,580	CSP	SIP	SRP	Sum Score	SenMayo
Age: Ref - Aged 55-64					
Aged 65-74	-0.05	0.02	0.06*	0.03	0.04
Aged 75-84	-0.09***	0.07***	0.13***	0.08***	0.11***
Aged 85+	-0.09***	0.05**	0.10***	0.05***	0.10***
Female	0.20***	-0.03	0.02	0.04	0.01
RE: Ref - Non-Hispanic White					
Non-Hispanic Black	0.12***	-0.02	-0.07***	-0.02	-0.07***
Hispanic	-0.01	-0.04*	-0.06**	-0.05**	-0.05*
Non-Hispanic Other	0.01	-0.02	-0.04*	-0.03	-0.03
Education: Ref - Less than High School					
High School	-0.02	0.00	0.02	0.00	0.03
Some College	0.00	-0.02	0.00	-0.01	0.01
College and Higher	-0.02	-0.04	-0.01	-0.03	0.02
BMI: Ref - Normal					
Overweight	0.01	0.02	-0.01	0.01	-0.03
Obese I	0.02	0.04	-0.02	0.02	-0.05*
Obese II	0.02	0.09***	0.06*	0.08***	0.00
Cumulative Packs Smoked	0.02	0.02	-0.01	0.01	-0.01
Total Drinks Weekly	0.02	0.01	0.03	0.02	0.04
Insomnia Symptoms	0.01	0.02	0.00	0.01	-0.01
Adjusted R2	0.203	0.404	0.247	0.342	0.291

Stars indicate significance based on FDR-adjusted p values. For highlighted cells, significance changed after FDR correction.

Predicting Aging-Related Health Outcomes - FDR Adjusted Version

Without DunedinPACE Added

Outcome	CSP			SIP			SRP			Sum Score			SenMayo			N
	beta/OR	p	p FDR	beta/OR	p	p FDR	beta/OR	p	p FDR	beta/OR	p	p FDR	beta/OR	p	p FDR	
PC GrimAge AA	0.00	0.920	0.920	0.31***	0.000	0.000	0.25***	0.000	0.000	0.29***	0.000	0.000	0.24***	0.000	0.000	3,580
PhenoAge AA	0.03	0.445	0.445	0.26***	0.000	0.000	0.21***	0.000	0.000	0.25***	0.000	0.000	0.21***	0.000	0.000	3,580
DunedinPACE	0.04	0.108	0.108	0.26***	0.000	0.000	0.19***	0.000	0.000	0.24***	0.000	0.000	0.16***	0.000	0.000	3,580
ExpBioAge AA	-0.01	0.692	0.692	0.23***	0.000	0.000	0.16***	0.000	0.000	0.20***	0.000	0.000	0.13***	0.000	0.000	2,660
Multimorbidity	0.00	0.925	0.925	0.15***	0.000	0.000	0.09***	0.000	0.000	0.12***	0.000	0.000	0.07***	0.001	0.001	3,580
Cognitive Functioning	-0.03	0.094	0.094	-0.07***	0.001	0.001	-0.05*	0.016	0.016	-0.07***	0.001	0.001	-0.05*	0.019	0.019	3,580
6yr Mortality	1.01	0.930	0.930	1.67***	0.000	0.000	1.38***	0.000	0.000	1.52***	0.000	0.000	1.32***	0.000	0.000	3,554

DunedinPACE Added as a Covariate

Outcome	CSP			SIP			SRP			Sum Score			SenMayo			N
	beta/OR	p	p FDR	beta/OR	p	p FDR	beta/OR	p	p FDR	beta/OR	p	p FDR	beta/OR	p	p FDR	
ExpBioAge AA	-0.02	0.410	0.410	0.16***	0.000	0.000	0.11***	0.000	0.000	0.13***	0.000	0.000	0.08**	0.002	0.002	2,660
Multimorbidity	-0.01	0.736	0.736	0.09***	0.000	0.000	0.04*	0.029	0.029	0.07**	0.004	0.004	0.03	0.097	0.097	3,580
Cognitive Functioning	-0.03	0.121	0.121	-0.06**	0.006	0.006	-0.04	0.066	0.066	-0.05**	0.008	0.008	-0.04	0.061	0.061	3,580
6yr Mortality	0.97	0.674	0.674	1.43***	0.000	0.000	1.22**	0.004	0.004	1.30***	0.001	0.001	1.18*	0.021	0.021	3,554

We did not apply multiple testing corrections globally throughout our study for the following three reasons:

(1) Hypothesis-Driven Nature of the Study: Our analyses were hypothesis-driven rather than exploratory. In the introduction, we explicitly stated that “cellular senescence is one of the molecular/cellular-level aging mechanisms that accumulate with advancing age and play an important pathogenic role in a number of adverse health outcomes.” We also highlighted that “a wide range of socio-demographic and behavioral factors have been linked to aging at the cellular/molecular level and thus could potentially be linked to cellular senescence as well.” Based on this prior evidence, we specified our key hypotheses at the end of the introduction: “We hypothesize that older ages, minority status, low SES, and risky health behaviors or conditions such as smoking, drinking, obesity, and sleep disorders will all be associated with higher levels of cellular senescence. We also hypothesize that the expression of senescence genes will be positively associated with age-related health outcomes.” These predefined hypotheses guided our analyses, reducing the risk of false positives associated with exploratory testing.

(2) Limited Number of Tests: For each socio-behavioral predictor (in models where each senescence score was the outcome) and for each health outcome (in models where each senescence score was the main predictor), we conducted only four models corresponding to the four senescence scores. We do not consider this to be a large number of tests that would necessitate multiple testing corrections.

(3) Commonly Accepted Practice in Similar Studies: Many existing studies with similar data sources, designs, and hypotheses to the current study do not use multiple testing corrections (e.g., Crimmins et al., 2021, 2024; Faul et al., 2023). This suggests that not

applying multiple testing corrections in hypothesis-driven, population-level studies of this kind is a commonly accepted practice.

8. Last point: Because these data types are so expensive, it can be understood easily why only one data set was used. However, it might be good if some crossvalidation analysis (as sensitivity analysis) is made. It would highlight if the findings are robust.

When conducting population health research, we really value the representativeness of the data source. We specifically selected the Health and Retirement Study (HRS) for this study due to its unique representativeness – it is representative of the community-dwelling older population in the United States. In addition to representativeness, it is also unique because it includes a comprehensive set of RNA data, DNA methylation (DNAm), flow cytometry, and clinical-level biomarkers collected from the same participants. Its focus on the older population guarantees the prevalence of health outcomes. This unique combination makes it particularly valuable for testing pre-defined gene lists in the context of aging-related health outcomes.

We agree that cross-validation or external validation is critical for studies aimed at developing new biomarkers. **However, we did not develop the biomarkers used in the current study. Instead we applied existing biomarkers, which have been validated in prior work.** In the field of population health research, many studies aim to apply previously validated biomarkers to specific datasets that offer unique features. For example, a recent study (Carroll et al., 2024) tested the SenMayo biomarker in a single dataset to evaluate its relevance for specific outcomes.

While other population-level data with RNA exist in the U.S., they have limitations that make them unsuitable for reliability testing. For example, NHANES (National Health and Nutrition Examination Survey) includes RNA and DNAm data and clinical biomarkers but lacks overlap between these biomarkers for the same participants. FHS (Framingham Heart Study) represents a specific region and racial-ethnic group, limiting its generalizability. MIDUS (Midlife in the United States) and Add Health (National Longitudinal Study of Adolescent to Adult Health) include younger populations with insufficient numbers of older adults, resulting in a low prevalence of aging-related health outcomes.

We recognize the importance of external validation and cross-country comparisons to enhance the robustness of findings. Nationally representative datasets from other countries, such as NICOLA (Northern Ireland Cohort for the Longitudinal Study of Aging) offer promising opportunity for such analyses. We are actively collaborating with this survey team as they finish their RNA development, and comparative projects are planned as part of our long-term research agenda. However, these efforts require substantial time and resources and are beyond the scope of this current study.

The following text has been added to the limitation section:

Our analyses reveal the potential of a set of cellular senescence scores as indicators of one of the hallmarks of aging, however, these measures should also be examined in different age groups and across different social contexts. At the current stage, conducting such comparative analyses across multiple comparable samples remains challenging. Future studies could potentially benefit from leveraging population-level multiomics data with similar representativeness.

Reviewer #2 (Remarks to the Author):

The paper by Wu et al. analyzes RNA/transcriptomics-based markers, namely CCA, MD, SASP, a summary senescence score, and SenMayo in the Health and Retirement Study and associates them with sociodemographic and behavioral factors and several age-related outcomes, including mortality, multimorbidity, biological age acceleration and epigenetic age acceleration. The find that the senescence scores mostly increased with age, were higher in women and individuals with class II obesity and associated with epigenetic aging, accelerated biological age, multimorbidity and mortality.

While I feel that the field definitely needs such new markers of biological aging that may provide a more comprehensive picture of aging and help us better understand the process, the work unfortunately suffers from several very fundamental flaws, such as a lack of correction for multiple testing and replication in an independent sample. There is also no description of the statistical analyses (an entire of two sentences of which the other says which program was used and the other refers to survey weights, which tells nothing about the analyses), so it is impossible to evaluate the statistical methods, let alone model parametrization and assumptions. I also feel that the paper suffers from lack of organization and clarity in the hypothesis and rationale. I hope my comments are helpful.

- In the title and throughout the text, I feel it is misleading to refer the scores to as “RNA-based Indicators of Cellular Senescence” as the scores are essentially gene expression profiles/transcript abundancies. More appropriate term, as used by the authors in the methods, is composite gene expression scores.

We revised the title and abstract based on the reviewer’s suggestion. We now refer to the scores as cellular senescence gene expression composite scores.

The new title is: **Gene Expression Composite Scores** of Cellular Senescence Predict Aging Health Outcomes in the Health and Retirement Study

The revised abstract is:

Cellular senescence, a hallmark of aging, can be quantified through the expression levels of genes involved in the canonical senescence pathway (CSP), senescence initiating pathway (SIP), and senescence response pathway (SRP). How cellular senescence links to sociodemographic characteristics, behavioral factors, and aging-related health outcomes in representative populations remains unknown. Using a nationally representative subsample from the U.S. Health and Retirement Study with RNA sequencing data, we calculated five cellular senescence gene expression composite scores: CSP, SIP, SRP, a summary senescence score, and SenMayo (based on the SenMayo gene list). Linear regression models assessed their associations with sociodemographic and behavioral factors (N=3,580), as well as aging-related health outcomes, including mortality (N=3,554), cognitive function (N=3,580), multimorbidity (N=3,580), biological age acceleration (N=2,660), and epigenetic age acceleration (N=3,580). Senescence scores increased with age ($\beta=0.04-0.13$, all $p<0.043$), except for CSP, which decreased ($\beta=-0.05$ to -0.09 , all $p<0.019$). Women ($\beta=0.04$, $p=0.021$) and individuals with class II obesity ($\beta=0.08$, $p<0.001$) exhibited higher senescence levels. All senescence scores, except CSP, were significantly associated with epigenetic aging, accelerated biological age, multimorbidity, cognitive function, and 6-year

mortality (all $p < 0.05$). These associations remained at least marginally significant after adjusting for DunedinPACE. Our results indicate that cellular senescence gene expression composite scores add to the explanation of health outcomes by epigenetic mechanisms.

- There are already several established transcript signatures associated with aging and/or aging-related traits, so it would be relevant to demonstrate to which extent the senescence-associated transcripts are represented in the existing signatures/scores. This would also help illustrate the added value of the senescence-related transcripts.

p16 and p21 are two of the most well-known and widely studied proteins in the field of cellular senescence. The genes that code these proteins – *CDKN2A* and *CDKN1A* – are included in the current gene lists. Peters et al. (2015) have created a transcriptomic age measure based on TWAS and the expression levels of 1497 genes were used to compute the measure. We compared the top 50 age-associated genes discovered by Peters et al. versus the gene lists included in the current study and found one overlapping with SIP (*MYC*) and one overlapping with SenMayo (*SERPINE2*). We would like to clarify that the gene lists used in our study were not developed by us but drawn from previously published literature. We believe that we included an appropriate selection of gene lists because they cover multiple key components of cellular senescence and have been previously tested/validated. We did not discover new transcripts/signatures/scores. Instead, we validated the gene lists that are specifically senescence-related at the population level. We also believe that, although cellular senescence is an important hallmark of biological aging, the cellular senescence gene expression composite scores do not aim at measuring biological aging in a comprehensive way. Measuring and understanding cellular senescence per se, distinct from overall aging, is also meaningful.

- The introduction is somewhat unfocused and long, presenting as list of topics that tap the tested exposures and outcomes, without a clear rationale or grounded hypothesis that would provide a foundation for the present work.

We have used an approach common in our field, and perhaps this differs from that expected by the reviewer. We have tried to restructure the current introduction to adequately motivate the research questions and hypotheses while making it more concise.

- To meet the aim stated by the authors “To understand how these composite scores are then associated with both potential upstream and downstream aging outcomes, they are related to multiple dimensions of aging – including accelerated epigenetic age, the multi-system physiological dysregulation indicated by accelerated Expanded Biological Age (ExpBioAge), multimorbidity, and mortality.” this work would have to be better structured, more rigorous and provide a solid rationale for the approach. Now it unfortunately feels seemingly random testing of all possible associations with variables that were available in the sample.

The aging-related outcomes used in the current study were not randomly selected. Instead, they were informed by the morbidity process model, which was introduced in the first paragraph of the introduction. Each outcome variable in our study corresponds to a specific dimension of the morbidity process model. These outcomes were selected and coded based on their relevance to the morbidity process model and their widespread use in the literature. The biological measures reflect different levels and aspects of biology.

- Related to the above, if the authors were to aim at comprehensive characterization of the senescence-associated gene expression scores, the paper would need a description of the statistical analyses, model specifications, tested assumptions, correction for multiple testing and most importantly, replication in an independent sample. All these are unfortunately lacking now. The only cues to the models can be obtained from the table headers and footnotes, which is certainly not enough to evaluate what has been done and how. As I understand from the tables, it seems that the models were only adjusted for technical covariates, so the associations (called “regressions” by the authors) are essentially bivariate correlations that are extremely prone to confounding. Therefore, the true nature and relevance of the results is hard to judge.

In the original manuscript, the statistical models were described in the Results section prior to presenting the results of each set of models. This formatting strategy was intended to avoid repetitiveness but now we recognize that it may have been unclear and confusing to readers. In the revised manuscript, we have included a complete description of the statistical analyses in the Statistical Analysis section of the Methods to ensure clarity and accessibility. Specifically:

Models Predicting Senescence Scores from Social and Behavioral Factors: We do not fit bivariate models, all social and behavioral factors were included in a single multivariate regression model, which also contained technical controls for batch/plate effects. This approach is now clearly stated in both the Results and Statistical Analysis sections of the manuscript. It is also clarified in the notes of all result tables.

Models Predicting Epigenetic Aging and Downstream Aging Outcomes from Senescence Scores: We do not test bivariate correlations, all models were adjusted for all social and behavioral factors. This approach is now clearly stated in both the Results and Statistical Analysis sections of the manuscript. It is also clarified in the notes of all result tables.

Revised statistical analysis section:

Statistical Analysis

First, to understand the sociodemographic and behavioral pattern of differences in senescence scores, social and behavioral variables were included together in the same OLS regression model to predict each senescence score, adjusted for technical covariates. Then, the associations between each senescence score and epigenetic aging measures are assessed in OLS regression models, adjusted for all social and behavioral factors and technical covariates. Then, the associations between each senescence score and each aging-related health outcome including ExpBioAge acceleration, multimorbidity, and cognitive function are assessed in OLS regression models, and the associations between senescence scores and 6-year mortality are assessed in logistic regression models, adjusted for all social and behavioral factors and technical covariates. Finally, DunedinPACE is included in all the models using the senescence score to predict aging-related outcomes as a covariate to see whether senescence scores explain additional variation to that explained by epigenetic aging in these outcomes. Survey weights for the HRS VBS innovative sub-sample (2016 VBS weight for analyses including DNA methylation and epigenetic clocks, telomeres, and homocysteine) are used to adjust for initial sample selection and differential consent and completion. All analyses are performed using Stata version 18.

For responses to concerns about multiple testing and replication in an independent sample, please refer to our responses to Reviewer 1's Comments 7 and 8.

A quick summary of our responses: We performed FDR corrections to all our models. The results did not change notably. We agree that cross-validation or external validation is critical for studies aimed at developing new biomarkers. However, we did not develop the biomarkers used in the current study. Instead, we applied existing biomarkers, which have been previously validated.

- As said above, I was very surprised to find that the data analysis section is only two sentences. Please provide a fully developed description of statistical analysis, which models were used, how the model assumptions were tested and met, how the models were parameterized and what was the rationale to define something as an exposure and something as an outcome. This is particularly relevant for analyses on sociodemographic and -economic factors and BMI. Which way around you expect the effect-outcome - sequence to be? Please also consider confounding and adjust for it.

We have updated the Statistical Analysis section to include necessary details. The rationale for defining exposures and outcomes in our study was explained in the Introduction section, where it is supported by both theoretical frameworks and empirical evidence.

Specifically for BMI, based on the framework outlined in the Introduction, we consider obesity to be a risk factor that could potentially trigger cellular senescence. So, BMI is treated as a predictor (exposure) in our models, rather than as an aging-related health outcome.

- A basic principle and ground rule of analyses like these, where a great number of associations is tested, is, after carefully adjusting for confounding, to perform correction for multiple testing, using FDR or Bonferroni correction. The associations that pass such correction are then usually replicated in an independent sample to determine robustness and generalizability. While I understand that not all associations found here can be tested in an independent sample, such testing should be performed at least for the main findings. I am also certain that after appropriate adjustment for confounding and multiple testing, the significant associations will become much fewer and easier to replicate. There are several publicly available datasets with transcriptomics or RNA-seq data, some including more detailed sample and covariate data linked, so these could serve as starting point for the replication. If not sufficient, perhaps collaboration with consortia or research groups in possession of such data is an option? Last resort option is to perform cross-validation.

For responses to concerns about multiple testing and replication in an independent sample, please refer to our responses to Reviewer 1's Comments 7 and 8.

A quick summary of our responses: We performed FDR corrections to all our models. The results did not change notably. We agree that cross-validation or external validation is critical for studies aimed at developing new biomarkers. However, we did not develop the biomarkers used in the current study. Instead we applied existing biomarkers, which have been validated in their own ways before (our study is a validation of those measures in some ways). These measures are not all available in a harmonized representative older sample that can be used for replication.

- Last sentence of the first paragraph in the result says “The average biological age of the sample is 68 years (SD=12 years), similar to the average chronological age by design.” Biological age can be measured in several different ways and using various scales and scores (epigenetic clocks are also biological ages!), so the reader would appreciate more detailed clarification in this context as to what is meant by biological age here (albeit the intro and methods provide some cues). I assume it is the ExpBioAge that you are referring to? If yes, I don’t think it is appropriate to call the ExpBioAge as *the* biological age, rather use the exact term and indicate which level of biological aging this taps into (cellular, organ, system-wide etc.)

We agree that this needs to be clarified. We now refer to ExpBioAge.

- Why only PhenoAge, GrimAge and DunedinPACE were considered among the epigenetic clocks? While older and trained only on chronological age, the Horvath and Hannum clocks would surely be worthwhile testing too.

The following paragraph and corresponding tables (below) have been added to the discussion section:

Second- and third-generation epigenetic clocks are more appropriate indicators of health-related biological aging because they focus on DNAm patterns associated with aging phenotypes, health risks, mortality, or health indicators instead of only chronological age. Empirically, second- and third-generation clocks have demonstrated better performance in terms of reflecting social determinants of health (e.g., SES) and predicting aging-related health outcomes (Crimmins et al., 2021, 2024; Faul et al., 2023). We limited the total number of clocks included in the analysis to avoid creating an unnecessarily large number of models. However, we have additionally examined the association between senescence scores and the two first-generation clocks – PC Horvath2 age acceleration and PC Hannam age acceleration (Higgins-Chen et al., 2022). As expected, the associations of senescence scores with PCHorvath2AA and PCHannumAA are generally weaker than the associations with the second- and third-generation clocks.

Senescence Scores Predicting Epigenetic Aging Measures Without Cell Type Controls

	PCGrimAgeAA	PhenoAgeAA	DunedinPACE	PCHorvath2AA	PCHannumAA
CSP	0.00	0.03	0.04	-0.03	-0.06*
SIP	0.31***	0.26***	0.26***	0.05	0.11***
SRP	0.25***	0.21***	0.19***	0.09***	0.15***
Sum Score	0.29***	0.25***	0.24***	0.07*	0.12***
SenMayo	0.24***	0.21***	0.16***	0.09***	0.15***

With Cell Type Controls

	PCGrimAgeAA	PhenoAgeAA	DunedinPACE	PCHorvath2AA	PCHannumAA
CSP	0.07***	0.06	0.09***	0.00	-0.02
SIP	0.09***	0.11***	0.13***	0.00	-0.02
SRP	0.05*	0.08**	0.05*	0.06*	0.05*

Sum Score	0.09***	0.12***	0.11***	0.04	0.02
SenMayo	0.01	0.05	-0.01	0.06*	0.05

Each cell is a separate model. All models are adjusted for all covariates and batch/plate.

- Also puzzles me a bit as to why the PC version was used only for GrimAge but not for the other clocks? Usually one uses the clock versions consistently and if inconsistencies are observed between the original and the PC versions, they are reported and discussed.

We did not use PCPhenoAge because PCPhenoAge was partially retrained on the HRS sample, the same sample that we are currently using (in the Higgins-Chen et al. paper, HRS was listed among the training sets in the methods section, and it also appears in supplemental table ST6, section 1D, as part of the training data for PCPhenoAge in addition to InCHIANTI.). No PC version was created for DunedinPACE but this clock has greater reliability than the original DunedinPoAm.

- Please clarify the assessment of the multimorbidity score, were these ever diagnoses or was a specific look-back window applies? Since this is based on self-report data, has any validation been made against diagnosis codes or established indices like CCI?

The multimorbidity score was based on HRS survey questions asking whether the respondent has been ever diagnosed with a set of specific disease/condition by healthcare professionals. Prior research has indicated that self-reported doctor's diagnosis works well in longitudinal surveys (Beckett et al., 2000). Although no validation has been made against diagnosis codes for the current study, prior research on HRS has shown that validity varies by condition (Cigolle et al., 2018). Consistent with conditions that showed the highest reliability for self-report (e.g., stroke, cancer, diabetes, lung disease, heart), we include those in the multimorbidity score, as opposed to including the conditions that tend to show more inconsistencies in reporting (e.g., hypertension, arthritis). Surveys including these conditions to index physical health are very commonly used in population health studies, with their public health relevance (Fisher et al., 2005 https://hrs.isr.umich.edu/sites/default/files/biblio/dr-009_0.pdf).

We acknowledge that future studies with a focus on diseases/conditions or mortality could leverage validated information sources such as health records. We added this as a limitation of the current study.

In the discussion, the authors acknowledge that adjusting for cell composition attenuated most of the associations to null and interpret this to be a result of "overadjustment". I find the interpretation very unusual as in such cases the attenuation usually indicates that it is the cell types that are (independently) associated with the outcomes, not the transcript levels. It is a very standard methodology for example in DNA methylation analysis to adjust for blood cell composition, as failing to do so can lead to spurious findings for the same reason. Please consider revising the interpretation.

We revised the corresponding discussion and we removed the statement claiming the inclusion of cell composition to be overadjustment. We think both the results with and without cell composition adjustment are meaningful and interesting, and thus include both results.

The fact that the main effect becomes attenuated after adjusting for cell types suggests that cell types are correlated with both the independent variable (cellular senescence) and the dependent variable (aging-related outcomes). A plausible explanation for this attenuation is that cellular senescence influences aging-related outcomes, at least partially, through altering the distribution of immune cells. This explanation is theoretically supported, as the Senescence-Associated Secretory Phenotype (SASP) can trigger changes in immune cell levels. Our primary aim is to identify whether a link exists between cellular senescence and aging-related outcomes, rather than to decompose this link into direct and indirect effects.

We acknowledged that adjusting for cell composition is a standard approach in analyses involving, for example, DNA methylation. This was the main motivation for performing the cell composition sensitivity analysis in our study. However, we believe the use of such an approach allows multiple interpretations of the meaning of results. While adjusting for cell composition can help better understand why DNA methylation or gene expression is linked to downstream physiological changes, the decision to apply this adjustment should depend on the research question. If the aim is to explore whether an association exists (as in the current study), the mediation effect through cell composition should not be interpreted as a spurious effect.

The revised paragraph in the discussion section regarding this issue is below:

Gene expression profiles can differ across cell types; therefore, we conducted sensitivity analyses to account for between-person variability in immune cell composition. A common approach is to adjust for DNA methylation-based estimates of major immune cell subsets (Reinius et al. 2012; Salas et al. 2018). However, the HRS includes flow cytometry data, which directly provides the percentages of these major immune cell subsets, including granulocytes, natural killer cells, B cells, CD4+ T cells, CD8+ T cells, and monocytes. When we controlled for these cell-type proportions, most of the age effects on senescence scores were no longer significant (Supplemental Table 6), and certain associations between senescence scores and aging-related health outcomes were attenuated or lost significance (Supplemental Tables 7 and 8). One plausible explanation for this attenuation is that immune cell distribution may lie on the causal pathway between the expression of cellular-senescence-related genes and the other variables of interest. For example, cellular senescence, in part through the SASP, can alter immune cell composition, which may subsequently affect aging-related health outcomes. Because our primary objective was to identify whether a link exists between cellular senescence and these variables—rather than to illustrate specific pathways or separate direct and indirect effects—the unadjusted model results remain most relevant to our aims. Nevertheless, these sensitivity analyses underscore the importance of clarifying how cellular senescence is triggered and how it influences health outcomes through other mechanisms such as immunosenescence.

- In the light of the above, the limitations-paragraph in the discussion is very much underdeveloped, but this can be remedied by addressing the main methodological and approach-related issues.

The revised limitation paragraph can be seen below:

The current study has limitations. First, our analyses support the potential of a set of cellular senescence scores as indicators of one of the hallmarks of aging, but these measures should also be examined across different demographic and social contexts. At the current

stage, conducting such comparative analyses across multiple comparable samples remains challenging. Future studies could potentially benefit from leveraging other population-level multi-omics datasets. Second, some aging-related health outcomes used in the current study rely on self or survivor reports, including multimorbidity and 6-year mortality. Although self-reports are commonly used in population surveys, future studies with a focus on diseases/conditions or mortality could leverage information sources such as health records. Third, representativeness is a strength of our sample, but it also reflects the sex and age structure of the population, especially the sex differences in survival to older ages. Specifically, the proportion of female participants in our study steadily increases with advancing age (e.g., 52.6% among those aged 55–64, 52.9% among those aged 65–74, 57.0% among those aged 75–84, and 65.9% among those aged 85+). Thus, a sample with more balanced sex ratios and a more evenly distributed age pattern could be more suitable for future studies that have a focus on the sex and age pattern of cellular senescence. Finally, this is a cross-sectional study, so no conclusions can be drawn regarding the causal role of senescence in any of the aging-related outcomes analyzed here.

- Would be very helpful if the authors provided a figure indicating the measurement occasions for collecting the data and obtaining the different measures.

The measurement occasions for data collection can be seen in the following table. We also revised the methods section to make sure the measurement occasion for each measure is clearly stated and added the table to the supplemental material.

2014	2016	2022
Systolic blood pressure, peak flow, and HbA1c for a random half of the HRS biomarker sample (Used to compute ExpBioAge)	Systolic blood pressure, peak flow, and HbA1c for a random half of the HRS biomarker sample (Used to compute ExpBioAge)	Vital status report (used to code 6-year mortality)
Height and weight for a random half of the HRS biomarker sample (Used to calculate BMI). When physical measures are not available/missing, self-reported height and weight on 2016 are used to calculate BMI.	19 clinical-level venous-blood-based biomarkers (Used to compute ExpBioAge)	
	Health outcomes, including cognitive function and multimorbidity	
	Epigenetic aging measures (PC GrimAge AA, PhenoAge, DunedinPACE) computed from venous-blood-based DNA methylation data	
	Gene expression composite scores (CSP, SIP, SRP, senescence summary, and SenMayo scores) computed	

from venous-blood-based RNA sequencing data

Height and weight for a random half of the HRS biomarker sample (Used to calculate BMI). When physical measures are not available/missing, self-reported height and weight on 2016 are used to calculate BMI.

All other covariates

Minor:

- In the abstract please specify what SenMayo refers to

SenMayo is not an abbreviation and hence cannot be further spelled out. But we have reworded to make it clearer.

- In the table footnotes, I believe you are referring to batch with “patch”?

We thank the reviewer for pointing out the typo. We have corrected it in the revised manuscript.

Reviewer #3 (Remarks to the Author):

Review of the Manuscript entitled “RNA-based Indicators of Cellular Senescence Predict Aging Health 2 Outcomes in the Health and Retirement Study ” by Wu Q and coauthors Briefly, using cross-sectional data from a large and nationally representative subsample of men and women, aged 56 years or older, from the U.S. Health and Retirement Study with available RNA sequencing data, the authors investigated the association between cellular senescence and age-related health outcomes. Five RNA-based cellular senescence scores (CCA, MD, SASP, a summary senescence score, and SenMayo) were calculated. Age related health outcomes included epigenetic age, the multi-system physiological dysregulation indicated by accelerated Expanded Biological Age (ExpBioAge), multimorbidity, and mortality. As result, senescence scores increased with age, except for CCA, which decreased. Women and individuals with class II obesity exhibited higher senescence levels. All senescence scores, except CCA, were significantly associated with epigenetic aging, accelerated biological age, multimorbidity, and 6- year mortality. Noteworthy, these associations remained significant after adjusting for GrimAge.

It is well-recognized that cellular senescence it a key mechanism in the patho-physiology of aging. As acknowledged in the introduction of the manuscript, previous studies showed a significant association between cellular senescence and age-related chronic conditions. Indeed, a point of strength of the analysis by Wu and coauthors is the large dimension of the sample population. Moreover, they used a comprehensive approach to measure cellular senescence. Regarding methodology, authors used an appropriate study design, valid and reliable measures and a correct analytic approach. Results are valid and robust, providing a significant contribution to the existing literature on this topic.

However, my comments are:

1. Contrary to what expected, CCA score was found to decrease with age and was not associated with age-related health outcomes. How this finding could be interpreted? I suggest to add a paragraph about this in the discussion.

We have added the following paragraph to the discussion section:

Though most senescence scores are higher among older age groups, the expression level of CSP genes is significantly lower (Table 2). In addition, unlike other senescence scores, the CSP score is not significantly associated with aging-related outcomes (Figure 2). These results imply that the genes included in the CSP list might capture an aspect of senescence which is not entirely detrimental. In fact, the capacity to induce CCA in response to stress and damage could be important and beneficial under certain circumstances, for example to prevent the proliferation of damaged cells and malignant transformation (Coppé et al. 2010). Choi and Lim (2011) found the induction of p21 is compromised in aging cells (the corresponding gene CDKN1A is included in the CSP list used in the current study). Previous studies also suggest that although CCA is considered a hallmark of cellular senescence, cell-cycle re-entry can occur under certain adverse circumstances particularly in tumor cells (Gorgoulis et al. 2019). Collectively, these previous works and our findings suggest that the effective activation of CCA could be compromised while aging. Relevantly, female sex is associated with higher CSP and senescence summary scores (Table 2). Since both SIP and SRP are not significantly associated with female sex, the elevated senescence summary score among females is likely mainly driven by CSP. Given the well-known lifespan advantage of females, the sex difference in CSP could be interpreted as the reduced capacity of CCA among males.

2. Multimorbidity included only 5 conditions: diabetes, cancer, chronic lung disease, heart disease and stroke. I would suggest to include a measure of cognitive decline among the age-related health outcomes to investigate whether cellular senescence is associated also with impair cognitive function in the sample population.

We thank the reviewer for this comment. We have added cognitive function as a separate aging-related health outcome, and updated the relevant text. The inclusion of cognitive function has the expected results. Senescence scores are associated with worse cognitive function.

Cognitive Function – Cognitive function is measured using a modified version of the Telephone Interview for Cognitive Status (TICS-m) in 2016. A cognitive function score ranging from 0-27 was generated by summing up the imputed TICS-m items provided in the HRS data, with higher values indicating better cognitive function.

Reviewer #4 (Remarks to the Author):

This descriptive study provides novel and valuable evidence that "senescence scores" derived from whole blood RNA sequencing (RNAseq) data can predict age-related health outcomes, including 6-year mortality, and demonstrate associations with socioeconomic and behavioral factors.

While descriptive, the study addresses an important gap in the field and highlights the potential utility of senescence biomarkers that can be obtained from blood from living humans in aging research. The manuscript is noteworthy, particularly because it validates the use of four out of five senescence scores in human whole blood cells—a setting where their utility has not been previously well-established. The findings could have significant implications for future human studies requiring reliable aging biomarkers, as well as preclinical research using animal models to evaluate geroprotective interventions for healthspan extension.

The manuscript presents a good case for publication and will be of interest to a broad audience of researchers working on aging and its associated health outcomes. However, I have some questions and suggestions.

Major Comments:

1. Renaming of Senescence Score Gene Lists:

The authors renamed three gene lists derived from Dehkordi et al. (2021): "Canonical Senescence Pathway" to "Cell Cycle Arrest (CCA)," "Senescence Initiators" to "Macromolecular Damage (MD)," and "Senescence Responses" to "Senescence-Associated Secretory Phenotype (SASP)." While the renaming is logical and perhaps enhances interpretability, it would be beneficial if the authors could explicitly clarify the rationale for these changes in the text. A brief explanation in the methods or discussion section would

provide transparency and perhaps help readers understand the biological implications from the named gene lists better.

We appreciate the reviewer's comment. To reduce confusion, we now name the gene lists using their original names: canonical senescence pathway (CSP), senescence initiating pathway (SIP), and senescence response pathway (SRP). In the main text, we still use MD, CCA, and SASP to explain the mechanisms of cellular senescence, but CSP, SIP, SRP are used to refer to the gene lists and the scores. We have clarified the relationship between these two sets of names in both the introduction and the methods sections.

2. Sex Distribution Across Age Groups:

The study benefits from an approximately equal proportion of sexes overall, but the distribution of males and females across the four age groups is unclear. Given the significant sex differences in aging and life expectancy (e.g., 76.2 years for males and 81.1 years for females in 2016 in United States, according to Natl Vital Stat Rep 2019 May;68(4):1-66), it is likely that later age groups (e.g., 75–84 and 85+) include a higher proportion of females. Providing sex-specific proportions for each age group would offer valuable context and help interpret the results.

We agree with the reviewer that understanding the sex distribution of each age group could help interpret the results. As expected, we found higher proportions of women among older age groups, as indicated by the following table and figure. We have included this as a limitation of the current study. The revised limitation section can be seen below:

The current study has limitations. First, our analyses support the potential of a set of cellular senescence scores as indicators of one of the hallmarks of aging, but these measures should also be examined across different demographic and social contexts. At the current stage, conducting such comparative analyses across multiple comparable samples remains challenging. Future studies could potentially benefit from leveraging other population-level multi-omics datasets. Second, some aging-related health outcomes used in the current study rely on self or survivor reports, including multimorbidity and 6-year mortality. Although self-reports are commonly used in population surveys, future studies with a focus on diseases/conditions or mortality could leverage information sources such as health records. Third, representativeness is a strength of our sample, but it also reflects the sex and age structure of the population, especially the sex differences in survival to older ages. Specifically, the proportion of female participants in our study steadily increases with advancing age (e.g., 52.6% among those aged 55–64, 52.9% among those aged 65–74, 57.0% among those aged 75–84, and 65.9% among those aged 85+). Thus, a sample with more balanced sex ratios and a more evenly distributed age pattern could be more suitable for future studies that have a focus on the sex and age pattern of cellular senescence. Finally, this is a cross-sectional study, so no conclusions can be drawn regarding the causal role of senescence in any of the aging-related outcomes analyzed here.

Age Group	Male (%)	Female (%)
55-64	47.4	52.6
65-74	47.1	52.9
75-84	43.0	57.0
85+	34.2	65.9

3. Sex-Specific Analysis:

As discussed briefly in ‘Discussion’ section in the manuscript, sex differences in aging are well-documented. Women tend to experience earlier onset of age-related conditions (particularly those driven by senescence and inflammation) but outlive men globally, regardless of socioeconomic status. This suggests that the aging process, including senescence progression, may differ between sexes. For instance, females may exhibit earlier associations between senescence scores and age-related conditions, while males could "catch up" at later ages. A bioinformatics study has shown there might be such a possibility: Ezra et al., doi: <https://doi.org/10.1101/2023.02.27.530179>.

To explore this, it would be insightful to test associations between senescence scores and age-related outcomes in a sex-specific and age-specific manner, rather than grouping all age ranges together with sex included as a covariate. It is possible that weak associations observed at younger ages become stronger at older ages because of converging senescence score trajectories between sexes. Even though this is a cross-sectional study and sample sizes for the 85+ group may be limited, the data could potentially reveal intriguing sex-specific patterns in the relationship between senescence scores and aging related health outcomes.

To investigate whether the relationship between cellular senescence and health outcome changes with age, and whether this “age-related change” itself differs by sex, we ran senescence-age interaction models separately for females and males. We ran these sex-stratified interaction models for four aging-related health outcomes - DunedinPACE, ExpBioAge AA, multimorbidity, and cognitive functioning. The b values and p values for the interaction terms can be seen below:

Outcome	CSP * Age		SIP * Age		SRP * Age		Sum Score * Age		SenMayo * Age		N
	b	p	b	p	b	p	b	p	b	p	
DunedinPACE (Female)	0.00	0.354	0.00	0.094	0.00	0.057	0.00	0.126	0.00	0.352	2,072
DunedinPACE (Male)	0.00	0.893	0.00	0.935	0.00	0.381	0.00	0.996	0.00	0.448	1,508
ExpBioAge AA (Female)	0.09	0.663	0.26*	0.039	0.31***	0.001	0.35*	0.014	0.36**	0.003	1,542
ExpBioAge AA (Male)	-0.21	0.212	-0.02	0.896	-0.04	0.764	-0.13	0.512	-0.01	0.939	1,118
Multimorbidity (Female)	-0.02	0.230	0.00	0.899	0.00	0.937	0.00	0.824	0.00	0.784	2,072
Multimorbidity (Male)	0.01	0.375	0.02*	0.044	0.00	0.622	0.02	0.275	0.01	0.590	1,508
Cognitive Functioning (Female)	0.00	0.925	-0.07	0.056	-0.08*	0.011	-0.09*	0.025	-0.10*	0.014	2,072

Cognitive Functioning (Male)	0.09	0.097	-0.06	0.156	-0.06	0.062	-0.05	0.236	-0.07	0.098	1,508
------	-------	-------	-------	-------	-------	-------	-------	-------	-------	-------

These are interesting results: The associations between senescence scores and outcomes sometimes differ with age – the senescence-ExpBioAge association becomes stronger with the increase of age only for females, and on the contrary, the senescence-cognition association becomes weaker with the increase of age only for females. These interactions seem to suggest that for females, the story is not consistent, and for males, no “catching up” can be observed. We believe these specific associations could be a future direction of research, but are not directly relevant to our current paper and do not change our general conclusions.

We hope the reviewer understands that a key challenge in this type of research is the need for a highly focused theoretical framework to guide hypothesis testing. Even for this manuscript in its current shape, where we explicitly stated that our hypotheses are grounded in the mortality-process model, another reviewer suggested that we were testing all possible associations without clear motivation. While we respectfully disagree with that critique, we also recognize that expanding our analyses to test additional hypotheses not directly tied to our main theoretical framework could challenge the validity of our work.

4. Ethnicity Distribution Across Sex and Age Groups:

The manuscript also examines differences across ethnic groups, but the distribution of sex and age within each ethnicity is not provided. Since both age (and sex) are key variables influencing senescence and health outcomes, understanding their distribution across the four ethnic groups is critical for interpreting the findings. It would be helpful to clarify whether differences in senescence scores between ethnicities might be partly driven by imbalances in sex or age distributions.

We thank the reviewer for the suggestion. We show how the sex ratio differs by age groups across racial/ethnic groups in the following figure. For all racial/ethnic groups except for non-Hispanic others, the proportions of women are lower overall in these older age groups, as expected.

We also show the relative proportions of age groups across racial/ethnic groups. The proportions in the youngest age group (age 55-64) among non-Hispanic Blacks, Hispanics, and non-Hispanic others are all significantly larger than that among non-Hispanic Whites. The proportions of age 75-84 among non-Hispanic Blacks, Hispanics, and non-Hispanic others are all significantly lower than that among non-Hispanic Whites.

However, we do not believe this could be used to explain the differences in senescence scores across racial/ethnic groups, since age has controlled as a covariate in our models. Nevertheless, we agree that the racial/ethnic difference in senescence should be further discussed. We believe that lower senescence level among minority groups is due to more “resilient” minorities survive to older ages. We added the following paragraph to the discussion section together with the table below:

In our sample, minority status is associated with lower levels of cellular senescence (i.e., lower SRP score for non-Hispanic Blacks; lower SIP, SRP, senescence summary, and SenMayo scores for Hispanics), which is opposite to our expectation. People who belong to racial/ethnic minority groups generally have more adverse exposures (e.g., environmental,

chemical, and psychological) and fewer protective resources (e.g., occupational, social, and healthcare) throughout their lifespan, and hence are hypothesized to have a higher level of senescence. The proportions in the youngest age group (aged 55-64) among racial/ethnic minority groups are all significantly larger than that among non-Hispanic Whites. The proportions of aged 75-84 among racial/ethnic minority groups are all significantly lower than that among non-Hispanic Whites, indicating differential survival across racial/ethnic groups (Supplemental Table 4). One explanation is that, in our analytical sample, people from racial/ethnic minority groups who survive to older age are more resilient and are healthier in the aspects that are relevant to the level of cellular senescence. However, such differential patterns of survival across racial/ethnic groups should be considered as the consequence of complex socioeconomic disparity rather than the influence of distinct underlying biological mechanisms.

Relative Proportions of Male vs Female across Racial/Ethnic Groups

Relative Proportions of Age Groups across Racial/Ethnic Groups

Age Group	Race/Ethnicity	Proportion	95% Conf. Interval		Wald Test P-Value
Age 55-64	NH White	38.4%	0.36	0.41	Ref
	NH Black	46.1%	0.41	0.52	0.011
	Hispanic	46.5%	0.40	0.53	0.016
	NH Others	57.4%	0.46	0.68	0.001
Age 65-74	NH White	35.2%	0.33	0.38	Ref
	NH Black	36.3%	0.31	0.42	0.718
	Hispanic	34.8%	0.29	0.41	0.910
	NH Others	32.4%	0.23	0.44	0.603
Age 75-84	NH White	18.9%	0.17	0.20	Ref
	NH Black	12.2%	0.10	0.15	0.000
	Hispanic	12.4%	0.10	0.16	0.000
	NH Others	7.5%	0.04	0.13	0.000
Age 85+	NH White	7.6%	0.07	0.09	Ref
	NH Black	5.5%	0.03	0.09	0.180
	Hispanic	6.3%	0.04	0.10	0.425

5. Combination of MD and SenMayo Scores:

Among the senescence scores, the MD (Macromolecular Damage) score appears to be the strongest predictor of age-related health outcomes. Would it be worthwhile to explore whether combining MD with the SenMayo score strengthens the associations? For example, if the combination improves predictive power, it may indicate complementary contributions of these gene lists. Conversely, if the association weakens, it could suggest that SenMayo includes genes less relevant to these specific outcomes, even if the gene list captures senescence itself. Testing such combinations could provide valuable insights into the relative contributions of each score and guide future refinements in senescence biomarkers.

We appreciate the reviewer's insightful comment. However, we want to clarify that the main purpose of the current study is not to develop new scores. The gene lists used in this study have already been validated, and their use here is to evaluate whether those validated scores are predictive in their original form. Though seeming to be out of the range of the current study, combining scores to optimize prediction of health outcomes is an excellent idea. It could be a promising future direction of research for studies dedicated to that specific mission.

We added this point to the discussion section:

Although our senescence scores were not developed to maximize predictive accuracy for clinical applications, their sensitivity to social and behavioral factors and strong associations with multiple dimensions of aging suggest significant potential for future research. These scores could serve as valuable tools for identifying mechanisms by which social and behavioral exposures influence aging biology. These scores may also aid in preclinical and clinical evaluations of interventions aimed at reducing the burden of cellular senescence. While clinical research plays a critical role in aging research, population-level survey data provide unique value by offering insights that are difficult to obtain in smaller, controlled clinical settings. Population-level studies, such as the Health and Retirement Study (HRS), enable the investigation of aging across diverse subpopulations, capturing the impact of social, economic, and behavioral factors on health outcomes. These data sources are also critical for understanding the interplay between biological and non-biological determinants of health, identifying disparities, and informing public health interventions. The HRS, with its nationally representative sample of older Americans, with extensive RNA-seq data and comprehensive collection of social, economic, behavioral, and health information, is a uniquely powerful resource for testing similar hypotheses and advancing translational research in aging. Maximizing prediction is not the aim of the current study, but future studies dedicated to that specific mission could consider combining the gene lists used in this study, or building on them to develop new algorithms.

Additional Comments:

- Clarity of Age-Related Associations:

The authors observe that associations between senescence scores and age-related outcomes are weaker in younger age groups and become stronger with increasing age. While this observation is consistent with the progressive nature of senescence, further discussion on the potential biological mechanisms driving this trend might enhance the manuscript. For

example, could this pattern reflect accumulating senescent cells with age or the increasing impact of senescence-related inflammation and macromolecular damage in older individuals?

In our models where each senescence score was used to predict aging-related health outcomes, age groups were included as covariates. However, no interactive effects between age group and senescence scores were assessed. Therefore, our results do not provide evidence on whether the associations between senescence scores and health outcomes differ by age group. While investigating such interactive effects could be an interesting avenue for future research, it is beyond the scope of our current study and deviates from our main theoretical framework. We believe that exploring these interactions requires a more focused, theory-driven hypothesis to ensure robust and meaningful findings.

- Future Applications in Translational Research:

The authors could expand on the potential translational applications of their findings. For example, how might these senescence scores be used in clinical settings to assess patient risk for age-related diseases or mortality? Additionally, the manuscript could discuss the implications for preclinical research for translation, such as how these scores might aid in evaluating geroprotective drugs or other interventions aimed at reducing senescence burden and/or increasing healthspan.

We thank the reviewer for this suggestion. The following paragraph has been added to the discussion section.

Although our senescence scores were not developed to maximize predictive accuracy for clinical applications, their sensitivity to social and behavioral factors and strong associations with multiple dimensions of aging suggest significant potential for future research. These scores could serve as valuable tools for identifying mechanisms by which social and behavioral exposures influence aging biology. These scores may also aid in preclinical and clinical evaluations of interventions aimed at reducing the burden of cellular senescence. While clinical research plays a critical role in aging research, population-level survey data provide unique value by offering insights that are difficult to obtain in smaller, controlled clinical settings. Population-level studies, such as the Health and Retirement Study (HRS), enable the investigation of aging across diverse subpopulations, capturing the impact of social, economic, and behavioral factors on health outcomes. These data sources are also critical for understanding the interplay between biological and non-biological determinants of health, identifying disparities, and informing public health interventions. The HRS, with its nationally representative sample of older Americans, with extensive RNA-seq data and comprehensive collection of social, economic, behavioral, and health information, is a uniquely powerful resource for testing similar hypotheses and advancing translational research in aging. Maximizing prediction is not the aim of the current study, but future studies dedicated to that specific mission could consider combining the gene lists used in this study, or building on them to develop new algorithms.

- Methodological Clarifications:

Were any steps taken to address potential confounding effects of medications, polypharmacy or comorbidities on senescence scores?

Medication use was not included as a covariate in our analyses. While the HRS dataset provides some information on medication use, it is limited to medications for certain chronic

diseases and contains a relatively high proportion of missing data. To preserve a sufficient sample size and maintain the robustness of our analyses, we decided not to include medication use as a covariate. Comorbidity was not included as a covariate because it is one of the primary outcomes of interest in this study, as defined by our theoretical framework.

References

- Beckett, M., Weinstein, M., Goldman, N., & Yu-Hsuan, L. (2000). Do Health Interview Surveys Yield Reliable Data on Chronic Illness among Older Respondents? *American Journal of Epidemiology*, *151*(3), 315–323. <https://doi.org/10.1093/oxfordjournals.aje.a010208>
- Carroll, J. E., Crespi, C. M., Cole, S., Ganz, P. A., Petersen, L., & Bower, J. E. (2024). Transcriptomic markers of biological aging in breast cancer survivors: A longitudinal study. *JNCI: Journal of the National Cancer Institute*, djae201. <https://doi.org/10.1093/jnci/djae201>
- Cigolle, C. T., Nagel, C. L., Blaum, C. S., Liang, J., & Quiñones, A. R. (2018). Inconsistency in the Self-report of Chronic Diseases in Panel Surveys: Developing an Adjudication Method for the Health and Retirement Study. *The Journals of Gerontology Series B: Psychological Sciences and Social Sciences*, *73*(5), 901–912. <https://doi.org/10.1093/geronb/gbw063>
- Crimmins, E. M., Klopach, E. T., & Kim, J. K. (2024). Generations of epigenetic clocks and their links to socioeconomic status in the Health and Retirement Study. *Epigenomics*, *16*(14), 1031–1042. <https://doi.org/10.1080/17501911.2024.2373682>
- Crimmins, E. M., Thyagarajan, B., Levine, M. E., Weir, D. R., & Faul, J. (2021). Associations of Age, Sex, Race/Ethnicity, and Education With 13 Epigenetic Clocks in a Nationally Representative U.S. Sample: The Health and Retirement Study. *The Journals of Gerontology: Series A*, *76*(6), 1117–1123. <https://doi.org/10.1093/gerona/glab016>
- Faul, J. D., Kim, J. K., Levine, M. E., Thyagarajan, B., Weir, D. R., & Crimmins, E. M. (2023). Epigenetic-based age acceleration in a representative sample of older Americans: Associations with aging-related morbidity and mortality. *Proceedings of the National Academy of Sciences*, *120*(9), e2215840120. <https://doi.org/10.1073/pnas.2215840120>
- Higgins-Chen, A. T., Thrush, K. L., Wang, Y., Minter, C. J., Kuo, P.-L., Wang, M., Niimi, P., Sturm, G., Lin, J., Moore, A. Z., Bandinelli, S., Vinkers, C. H., Vermetten, E., Rutten, B. P. F., Geuze, E., Okhuijsen-Pfeifer, C., van der Horst, M. Z., Schreiter, S., Gutwinski, S., ... Levine, M. E. (2022). A computational solution for bolstering reliability of epigenetic clocks: Implications for clinical trials and longitudinal tracking. *Nature Aging*, *2*(7), 644–661. <https://doi.org/10.1038/s43587-022-00248-2>
- López-Otín, C., Blasco, M. A., Partridge, L., Serrano, M., & Kroemer, G. (2023). Hallmarks of aging: An expanding universe. *Cell*, *186*(2), 243–278. <https://doi.org/10.1016/j.cell.2022.11.001>

Marttila, S., Rajić, S., Ciantar, J., Mak, J. K. L., Junttila, I. S., Kummola, L., Hägg, S., Raitoharju, E., & Kananen, L. (2024). Biological aging of different blood cell types. *GeroScience*. <https://doi.org/10.1007/s11357-024-01287-w>

Peters, M. J., Joehanes, R., Pilling, L. C., Schurmann, C., Conneely, K. N., Powell, J., Reinmaa, E., Sutphin, G. L., Zhernakova, A., Schramm, K., Wilson, Y. A., Kobes, S., Tukiainen, T., Ramos, Y. F., Göring, H. H. H., Fornage, M., Liu, Y., Gharib, S. A., Stranger, B. E., ... Johnson, A. D. (2015). The transcriptional landscape of age in human peripheral blood. *Nature Communications*, 6(1), 8570. <https://doi.org/10.1038/ncomms9570>

Zhang, Z., Reynolds, S. R., Stolrow, H. G., Chen, J.-Q., Christensen, B. C., & Salas, L. A. (2024). Deciphering the role of immune cell composition in epigenetic age acceleration: Insights from cell-type deconvolution applied to human blood epigenetic clocks. *Aging Cell*, 23(3), e14071. <https://doi.org/10.1111/acel.14071>

Remarks from the Reviewer:

“The manuscript has improved for several parts in the revision. However, my foremost concern, replication in an independent sample or cross-validation as a bare minimum, remains unaddressed.

Regarding replication, the authors state that “We agree that cross-validation or external validation is critical for studies aimed at developing new biomarkers. However, we did not develop the biomarkers used in the current study. Instead we applied existing biomarkers, which have been validated in their own ways before (our study is a validation of those measures in some ways). These measures are not all available in a harmonized representative older sample that can be used for replication”.

As I specified in my initial comment (and as acknowledged by the authors themselves), replication (or cross-validation) is typically considered an essential component of biomarker validation, rather than a separate or optional downstream step. While the biomarkers applied in the current study have been previously developed, they have not been validated (replicated) in the context of the predictions presented here. The authors should therefore consider to seek replication in an independent sample or, where that is not feasible, to implement cross-validation strategies within the current dataset.”

Response to the Reviewer:

We appreciate the reviewer’s suggestion to conduct cross-validation (CV), and a CV has been done. However, we respectfully note that no training, model fitting, or parameter tuning was performed in this study. The gene expression composite scores we applied, which are basically the average expression level of genes, were constructed based on existing gene lists from the literature and were not derived or optimized using the current dataset. Typically this type of application without a specific “training” process does not require additional validation, but as an extra robustness check, we conducted a 5-fold cross-validation for all models linking senescence scores to aging outcomes. The average CV R^2 values, alongside the R^2 values from the full-sample models, can be seen in the table below. As expected, CV R^2 values were slightly lower than those from the full-sample models, suggesting that the explanatory power of the models may be somewhat reduced when applied to other populations.

As the reviewer noted, while the gene lists we used have been previously validated, their associations with aging-related health outcomes were evaluated for the first time in our study. We acknowledge that the credibility of these associations depends heavily on the quality and representativeness of the dataset. We believe that the Health and Retirement Study (HRS) provides an excellent resource for this purpose, given its large, population-representative sample, high-quality multi-omics data, and comprehensive aging phenotyping. We agree that it would be interesting to examine whether these gene expression composite scores are associated with aging-related health outcomes in another dataset. However, we also note that such an analysis would only be meaningful if the external dataset is highly comparable in terms of sample characteristics, gene expression profiling protocols, and health outcome measures. Without such

comparability, differences in associations may reflect underlying population or measurement differences rather than the reproducibility of associations. That said, exploring how associations vary across populations is itself an important research direction, though we consider that a distinct question beyond the scope of the current study and should be examined in the future.

We think our first limitation point in the current manuscript clearly discussed this future direction: “First, our analyses support the potential of a set of cellular senescence scores as indicators of one of the hallmarks of aging, but these measures should also be examined across different demographic and social contexts. At the current stage, conducting such comparative analyses across multiple comparable samples remains challenging. Future studies could potentially benefit from leveraging other population-level multi-omics datasets.”

We included the CV result table in the supplemental materials together with the following notes:

“The gene expression composite scores we applied were constructed based on existing gene lists from the literature and were not derived or optimized using the current dataset. No training, model fitting, or parameter tuning was performed in this study. However, as an extra robustness check, we conducted a 5-fold cross-validation for all models linking senescence scores to aging outcomes. The average CV R^2 values alongside the R^2 values from the full-sample models are reported. As expected, CV R^2 values were slightly lower than those from the full-sample models, suggesting that the explanatory power of the models may be somewhat reduced when applied to other populations.”

Supplemental Table 15. Five-Fold Cross-Validation Results:

Outcome	Predictor	CV R2	Main Model R2	RMSE	MAE
PCGrimAgeAA	CSP	0.31	0.35	3.26	2.53
PCGrimAgeAA	SIP	0.37	0.41	3.12	2.43
PCGrimAgeAA	SRP	0.35	0.40	3.17	2.46
PCGrimAgeAA	Sum Score	0.36	0.40	3.15	2.45
PCGrimAgeAA	SenMayo	0.35	0.39	3.17	2.46
PhenoAgeAA	CSP	0.01	0.04	6.82	5.22
PhenoAgeAA	SIP	0.04	0.08	6.68	5.10
PhenoAgeAA	SRP	0.04	0.07	6.73	5.14
PhenoAgeAA	Sum Score	0.04	0.08	6.70	5.11
PhenoAgeAA	SenMayo	0.04	0.07	6.72	5.12
DunedinPACE	CSP	0.18	0.22	0.13	0.10
DunedinPACE	SIP	0.22	0.26	0.13	0.10
DunedinPACE	SRP	0.20	0.25	0.13	0.10
DunedinPACE	Sum Score	0.21	0.26	0.13	0.10
DunedinPACE	SenMayo	0.19	0.24	0.13	0.10
ExpBioAgeAA	CSP	0.05	0.09	8.31	6.08
ExpBioAgeAA	SIP	0.08	0.12	8.16	6.01
ExpBioAgeAA	SRP	0.07	0.11	8.22	6.04
ExpBioAgeAA	Sum Score	0.07	0.11	8.18	6.03

ExpBioAgeAA	SenMayo	0.05	0.10	8.29	6.10
Multimorbidity	CSP	0.09	0.13	0.96	0.76
Multimorbidity	SIP	0.11	0.14	0.95	0.75
Multimorbidity	SRP	0.10	0.14	0.95	0.76
Multimorbidity	Sum Score	0.11	0.14	0.95	0.75
Multimorbidity	SenMayo	0.10	0.13	0.95	0.75
Cognitive Function	CSP	0.26	0.29	3.79	2.99
Cognitive Function	SIP	0.26	0.29	3.78	2.98
Cognitive Function	SRP	0.26	0.29	3.79	2.99
Cognitive Function	Sum Score	0.26	0.29	3.79	2.99
Cognitive Function	SenMayo	0.26	0.29	3.79	2.99
6yrMortality	CSP	0.20	0.24	0.52	0.27
6yrMortality	SIP	0.15	0.26	0.47	0.22
6yrMortality	SRP	0.05	0.25	0.46	0.21
6yrMortality	Sum Score	0.12	0.25	0.50	0.25
6yrMortality	SenMayo	0.22	0.25	0.50	0.26

Note,

CV R², RMSE, and MAE values in the table are the average values across the 5 folds.

For logistic regression models (models using 6-year mortality as the outcome), pseudo-R-squares (McFadden's R-squares) are calculated.

The gene expression composite scores we applied were constructed based on existing gene lists from the literature and were not derived or optimized using the current dataset. No training, model fitting, or parameter tuning was performed in this study. However, as an extra robustness check, we conducted a 5-fold cross-validation for all models linking senescence scores to aging outcomes. The average CV R² values alongside the R² values from the full-sample models are reported. As expected, CV R² values were slightly lower than those from the full-sample models, suggesting that the explanatory power of the models may be somewhat reduced when applied to other populations.

Remarks from the Reviewer:

“The inclusion of cross-validation analysis is a significant improvement. However, the results are not adequately addressed—they are only presented in the supplementary materials and not discussed at all in the main text. This omission is problematic, particularly given the cross-validation results for 6-year mortality. The CV R^2 values are notably lower than the main model R^2 , with drops exceeding 0.10, which commonly suggest overfitting. This may indicate that the model is overly complex or capturing noise rather than true signal. Similarly, the RMSE/MAE ratios exceeding 1.5 point to substantial variance in prediction errors, possibly due to outliers. These issues should be openly discussed in the manuscript to provide a transparent interpretation of the findings and their implications.

Moreover, the authors state that “The gene expression composite scores we applied were constructed based on existing gene lists from the literature and were not derived or optimized using the current dataset. No training, model fitting, or parameter tuning was performed in this study.” This statement is not entirely accurate. While the individual genes may have been previously associated with aging-related outcomes, the composite signature developed here is new. Whenever a new predictive combination is created—whether labelled a “signature” or “profile”—its performance needs to be rigorously assessed. The best practice is validation in an independent cohort, or if that is not feasible, thorough internal validation such as cross-validation.

If the authors wish to maintain the claim that no training or model fitting was performed, they should acknowledge that this limits the robustness of their approach.

Finally, the absence of replication in an independent cohort should be clearly stated as a limitation of the study.”

Response to the Reviewer:

We thank the reviewer for the detailed recommendation, and we have revised the main text accordingly - We revised the first point of the limitation section and made it a separate paragraph to discuss the issue of replication/validation and the CV results.

Specifically, the revised limitation section can be seen below:

The current study has limitations. First, although the gene expression composite scores applied here were constructed using pre-defined gene lists from the literature and not derived or optimized using the current dataset, it remains important to rigorously evaluate their predictive performance on the selected outcomes, ideally in an independent cohort comparable to the HRS. However, the HRS is unique in its combination of a large, population-representative sample, high-quality multi-omics data, and comprehensive aging phenotyping. At present, accessing a comparable dataset for external validation is challenging. As an alternative, we conducted 5-fold cross-validation (CV) for all models linking senescence scores to aging outcomes. The average CV R^2 values, full-sample R^2 values, RMSE, and MAE are reported in **Supplemental Table 13**. In general, CV R^2 values were slightly lower than those from the full-sample models, indicating that model explanatory power may decline modestly when applied to new populations. The drop in R^2 values and the ratio of RMSE and MAE were particularly notable in models predicting 6-

year mortality, suggesting potential overfitting in that context. As more population-level multi-omics datasets become available, further assessment of the generalizability and predictive performance of these composite scores will be possible.

Second, some aging-related health outcomes used in the current study rely on self or survivor reports, including multimorbidity and 6-year mortality. Although self-reports are commonly used in population surveys, future studies with a focus on diseases/conditions or mortality could leverage information sources such as health records. Third, representativeness is a strength of our sample, but it also reflects the sex and age structure of the population, especially the sex differences in survival to older ages. Specifically, the proportion of female participants in our study steadily increases with advancing age (e.g., 52.6% among those aged 55–64, 52.9% among those aged 65–74, 57.0% among those aged 75–84, and 65.9% among those aged 85+). Thus, a sample with more balanced sex ratios and a more evenly distributed age pattern could be more suitable for future studies that have a focus on the sex and age pattern of cellular senescence. Finally, this is a cross-sectional study, so no conclusions can be drawn regarding the causal role of senescence in any of the aging-related outcomes analyzed here.